# Isoform-specific roles for AKT in affective behavior, spatial memory, and extinction related to psychiatric disorders

Helen Wong[1†]*, Josien Levenga[1,2†], Lauren LaPlante[1], Bailey Keller[1], Andrew Cooper-Sansone[1], Curtis Borski[1], Ryan Milstead[3], Marissa Ehringer[1,3], Charles Hoeffer[1,2,3]*

[1]Institute for Behavioral Genetics, University of Colorado, Boulder, United States; [2]Linda Crnic Institute, Anschutz Medical Center, Aurora, United States; [3]Department of Integrative Physiology, University of Colorado, Boulder, United States

**Abstract** AKT is implicated in neurological disorders. AKT has three isoforms, AKT1/AKT2/AKT3, with brain cell type-specific expression that may differentially influence behavior. Therefore, we examined single *Akt* isoform, conditional brain-specific *Akt1*, and double *Akt1/3* mutant mice in behaviors relevant to neuropsychiatric disorders. Because sex is a determinant of these disorders but poorly understood, sex was an experimental variable in our design. Our studies revealed AKT isoform- and sex-specific effects on anxiety, spatial and contextual memory, and fear extinction. In *Akt1* mutant males, viral-mediated AKT1 restoration in the prefrontal cortex rescued extinction phenotypes. We identified a novel role for AKT2 and overlapping roles for AKT1 and AKT3 in long-term memory. Finally, we found that sex-specific behavior effects were not mediated by AKT expression or activation differences between sexes. These results highlight sex as a biological variable and isoform- or cell type-specific AKT signaling as potential targets for improving treatment of neuropsychiatric disorders.

*For correspondence:
hw460@nyu.edu (HW);
charles.hoeffer@colorado.edu
(CH)

†These authors contributed
equally to this work

Competing interests: The
authors declare that no
competing interests exist.

Reviewing editor: Mary Kay
Lobo, University of Maryland,
United States

## Introduction

Psychiatric disorders, including schizophrenia, major depressive disorder (MDD), and anxiety, are important public health burdens with large societal and economic costs (*Wittchen et al., 2011*; *Vigo et al., 2016*). Genetics is known to play a significant role in the manifestation of psychiatric illnesses (*Sullivan et al., 2003*; *Schumacher et al., 2011*; *Musci et al., 2019*). Numerous studies in humans and experimental model systems have identified genetic variations that can promote abnormal neural function underlying the occurrence of these disorders (*Renoir, 2014*). Detailed molecular studies have also led to the development of many pharmacological treatments for psychiatric illnesses. However, the effectiveness of available therapies is still limited and many patients remain untreated (*Gould et al., 2007*; *Duman and Voleti, 2012*). This may be due in part to lack of information about the specificity of neuromolecular signaling pathways involved in the manifestation of individual behaviors and processes associated with the symptomology of psychiatric disorders as well the specific signaling effects within different neural cell types and those impacted by sex. Consequently, further identification of critical signaling pathways are necessary for developing new therapeutic targets and improving efficacy of existing ones.

The protein kinase B (PKB/AKT) family of serine/threonine kinases is involved in numerous neuromolecular signaling processes (*Kandel and Hay, 1999*; *Franke, 2008*) and has been implicated in neurological and psychiatric disorders (*Saudou et al., 1998*; *Ikeda et al., 2004*; *Griffin et al., 2005*). For example, *Akt1* haplotypes have been identified in schizophrenic patients (*Emamian et al., 2004*;

*Ikeda et al., 2004*; *Schwab et al., 2005*; *Xu et al., 2007*). Additionally, antidepressants, antipsychotics, and mood stabilizers are known to modify AKT activity (*Li et al., 2004*; *Krishnan et al., 2008*; *Beaulieu et al., 2009*; *Li et al., 2010*; *Park et al., 2014*). AKT is expressed as three isoforms termed AKT1/PKBα, AKT2/PKBβ, and AKT3/PKBγ in the brain. Each isoform shows significant homology with one another and across species, being highly conserved in both humans and mice. Despite this homology, the isoforms exhibit different expression patterns in the brain and regulate the expression of synaptic plasticity differently (*Levenga et al., 2017*). Therefore, it is possible that the different isoforms also contribute differentially to cognitive processes affected in schizophrenia and affective disorders, such as memory formation and extinction. Unfortunately, the overwhelming number of human and animal studies have investigated AKT from a general perspective not delineating between isoforms. A few single AKT isoform studies have been performed with conflicting results (*Chang et al., 2016*; *Wang et al., 2017*). Additionally, no study to date has simultaneously examined AKT isoform deficiencies using a comprehensive battery of behavior tests. Finally, the few studies that have examined single AKT isoforms have ignored sex as a variable and, therefore, may have missed important sex-linked effects.

To address these gaps in our knowledge of AKT signaling related to psychiatric disorders, we selectively removed AKT isoforms from the mouse brain using complementary genetic approaches, generating mutant mice with single-isoform deficiencies in AKT1, AKT2, or AKT3; selective loss of AKT1 in forebrain excitatory neurons; or double loss of AKT1 and AKT3 isoforms. With these mice, we performed murine behavioral assays that are models for affective behavior, spatial learning and memory, and associative fear memory and extinction. The behavioral results provide multiple lines of evidence demonstrating specific and overlapping roles for AKT isoforms in the activity of neural circuits that may model brain processes impacted in neuropsychiatric disorders. On balance, AKT1 exerts the strongest individual isoform effects on cognition and behavior, influencing males selectively in many of the tests we performed. Like AKT1, AKT2 affects anxiety-like behavior in a sex-specific fashion and impacts contextual memory. AKT3 exerted no effects on its own but in the background of AKT1 deficiency, enhanced deficits we observed in *Akt1* mutants. Therefore, this study provides novel evidence demonstrating AKT isoform-specific regulation of neural function in a sex-specific fashion. Combined with the fact that AKT isoforms are expressed in a cell type-specific fashion, these findings improve understanding of how AKT activity is specifically involved in distinct behaviors and neurobiological processes relevant to diagnoses and therapies directed against AKT signaling pathways for treating psychiatric disorders.

## Results

### *Akt* deficiency alters anxiety-like behavior in an isoform- and sex-specific manner

To investigate the isoform-specific contributions of AKT and any sex-specific effects on cognitive processes, we began all behavioral studies of *Akt* mutant mice with a basic locomotor assessment using the open field arena (OFA) test. Besides locomotor activity, OFA assesses anxiety-like behavior by measuring time spent exploring the periphery or exposed center of the arena (*Bouwknecht and Paylor, 2008*). OFA testing revealed that male *Akt1* KO mice spend significantly less time exploring the center compared with wild-type (WT) littermate controls (*Figure 1A*; t(42)=4.789, p<0.001), indicating increased anxiety-like behavior, which was not due to differences in overall movement (*Figure 1A*). In contrast, female *Akt1* WT and KO mice were indistinguishable (*Figure 1B*). No genotype effects were observed in *Akt2* or *Akt3* mutants for both sexes (*Figure 1A,B*).

To complement our OFA studies, we also performed elevated plus maze (EPM) testing, which examines anxiety-like behavior and exploratory drive in rodents (*Bouwknecht and Paylor, 2008*). Time spent exploring the EPM open (exposed) arms is a measure of anxiolytic-like behavior. Consistent with increased anxiety-like behavior, *Akt1* mutants also displayed a male-specific reduction in open arm time compared with controls (*Figure 1C*; t(42)=2.156, p=0.037), which was not due to differences in distance moved (*Figure 1C*). EPM testing also revealed decreased open arm time in *Akt2* mutant males compared with WT controls (*Figure 1C*; t(28)=2.651, p=0.013) that was not due to differences in overall movement (*Figure 1C*), indicating increased anxiety-like behavior. In contrast, *Akt1* or *Akt2* deletion had no effect on female mice (*Figure 1D*) while *Akt3* deletion had no

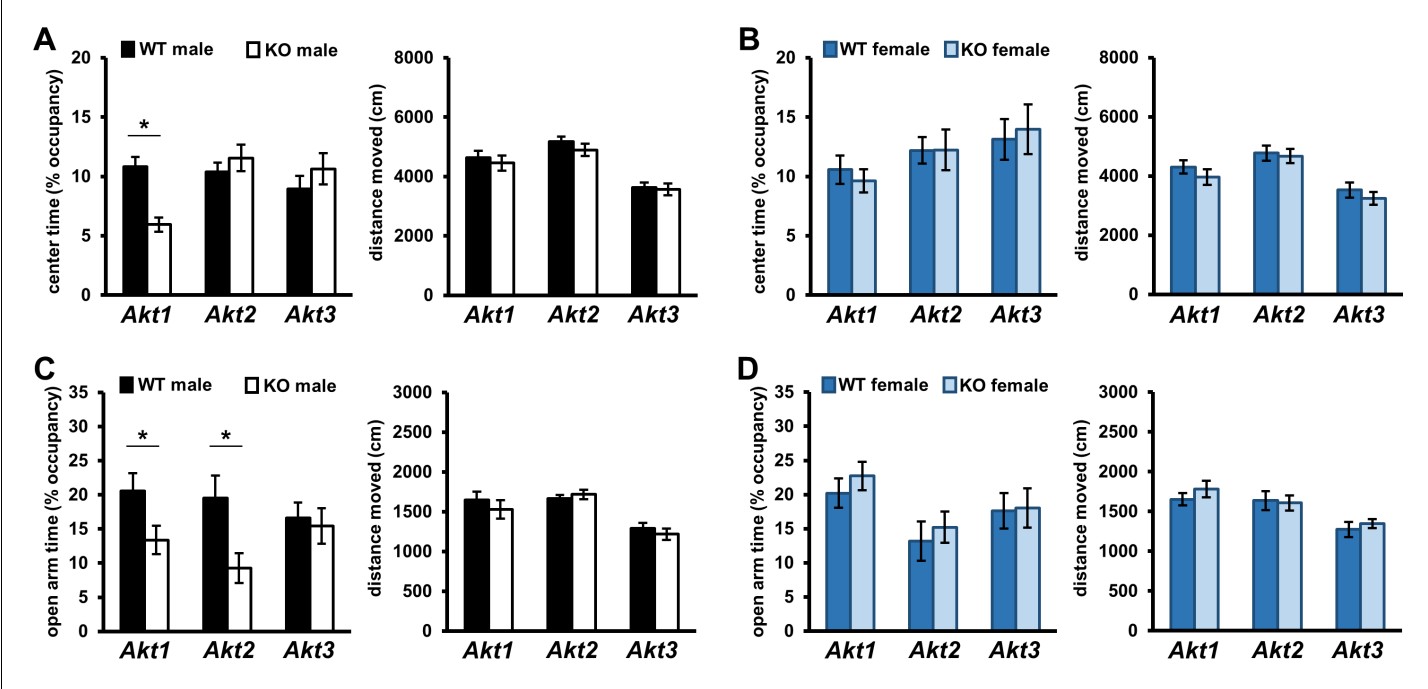

**Figure 1.** *Akt* deficiency affects the expression of anxiety-related behavior in an isoform- and sex-specific fashion. Male and female mice with single-isoform deletions of *Akt1*, *Akt2*, or *Akt3* were assessed for anxiety-related behaviors in the open field arena (OFA) and elevated plus maze (EPM) tests. (A–B) OFA activity shown as percent time spent exploring the center zone out of total arena time in *Akt* mutant (A) male and (B) female mice. *Akt1* KO male mice show reduced center time compared to their WT controls. (C–D) EPM activity shown as percent time spent in the open arms out of total maze time in *Akt* mutant (C) male and (D) female mice. *Akt1* KO and *Akt2* KO male mice show reduced open arm time compared with their respective WT controls. Differences in OFA center or EPM open arm times are not explained by locomotor alterations between KO and WT mice. No differences between female KO and WT mice were observed. *p<0.05. N = *Akt1* (WT-M = 22, KO-M = 22, WT-F = 16, KO-F = 13); *Akt2* (WT-M = 13–14, KO-M = 15–16, WT-F = 12–13, KO-F = 11); *Akt3* (WT-M = 19, KO-M = 23, WT-F = 17–18, KO-F = 18).

The online version of this article includes the following source data and figure supplement(s) for figure 1:

**Source data 1.** Figure 1 source data.
**Figure supplement 1.** Schematic representation of the experimental timeline.

effect on either sex (*Figure 1C,D*) in the EPM. Together these OFA and EPM data demonstrate that *Akt1* and *Akt2* deficiency affect the expression of anxiety-related behaviors in a sex-specific fashion.

## AKT1 deficiency leads to a mild spatial memory impairment

We next examined isoform-specific contributions of *Akt* to spatial learning and memory using the Morris Water Maze (MWM) (*Morris, 1984*). In this assay, mice are trained to find a hidden escape platform in a pool of water using visual cues. Spatial learning is measured by the escape latency curve over the training period. Cognitive flexibility also can be tested in this assay with reversal training, in which mice are re-trained to find the platform in the quadrant of the maze opposite the original training location. Finally, intact vision, required for this task, is assessed in a visible platform training component. No latency differences in each training component were detected for any *Akt* mutants compared to WT controls (*Figure 2*), indicating normal spatial learning, reversal learning, and visual acuity. After MWM training, spatial memory efficacy is evaluated using a probe test in which the escape platform is removed, and time spent in the target quadrant where the platform had been located and frequency of crossing the specific platform location are measured. We found no differences in target quadrant time between *Akt1* KO and WT mice (*Figure 2A,B*; males t(16) =1.187, p>0.05; females t(16)=-.489, p>0.05), but *Akt1* KO males displayed reduced target platform crossings compared with controls (*Figure 2A*; t(16)=2.389, p=0.030), indicating a sex-specific deficit in more precise spatial memory. Probe tests revealed no differences in *Akt2* KO (*Figure 2C,D*) or

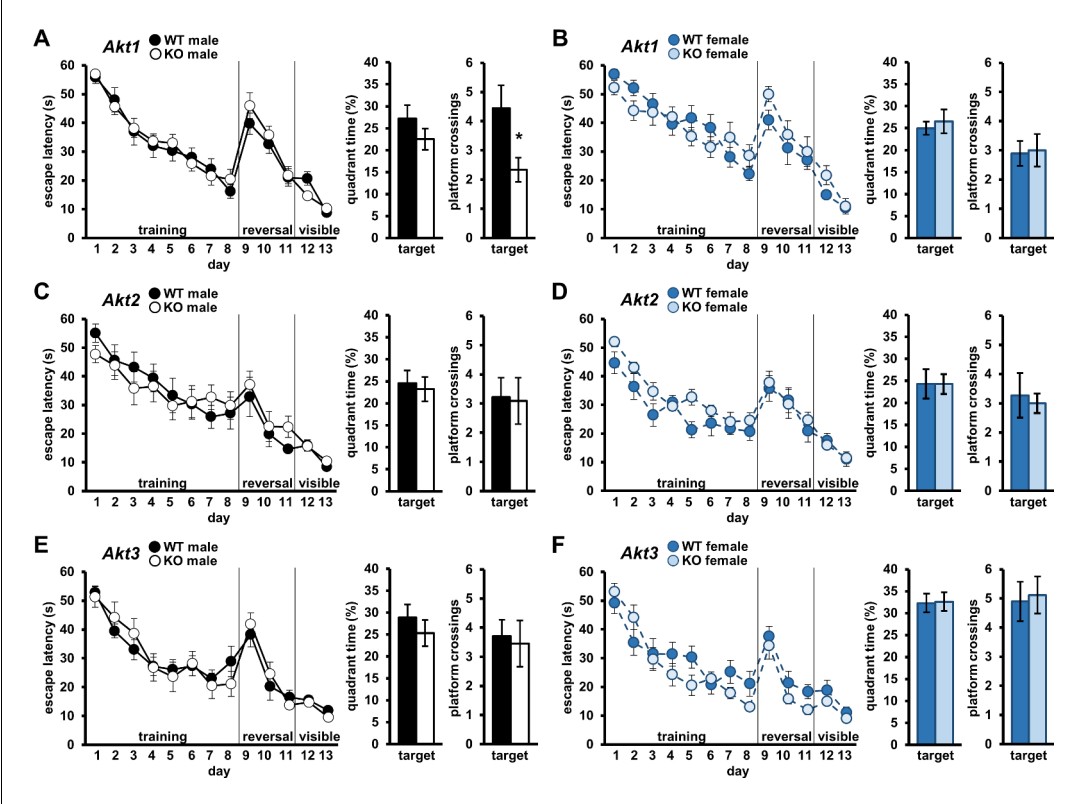

**Figure 2.** *Akt* deficiency affects spatial memory formation in an isoform- and sex-specific fashion. Spatial learning and memory were assessed using the Morris Water Maze (MWM) in male and female mice with single-isoform deletions of *Akt1, Akt2,* or *Akt3. Left graph*: Learning measured by latency of mice to escape the MWM during training to locate a hidden escape platform (days 1–8), reversal training to locate the hidden escape platform moved to the opposite quadrant (days 9–11), and visible platform training to locate the escape platform marked by a visual cue (days 12–13). Memory measured by percent time spent in the target quadrant where mice were trained to locate the hidden escape (*left bar graph*) and number of platform location crossings in the target quadrant (*right bar graph*) during a probe test. MWM performance in (A) *Akt1* KO male, (B) *Akt1* KO female, (C) *Akt2* KO male, (D) *Akt2* KO female, (E) *Akt3* KO male, and (F) *Akt3* KO female mice. *Akt1* KO male mice showed a significant reduction in platform crossings compared with WT controls during the probe test. No other differences were detected between *Akt* isoform KO and WT mice. *p<0.05. N = *Akt1* (WT-M = 9 KO-M = 9, WT-F = 9, KO-F = 9); *Akt2* (WT-M = 9, KO-M = 11, WT-F = 11, KO-F = 9); *Akt3* (WT-M = 10, KO-M = 11, WT-F = 10, KO-F = 9). The online version of this article includes the following source data for figure 2:

**Source data 1.** Figure 2 source data.

*Akt3* KO mice (*Figure 2E,F*). Combined, these data suggest a subtle requirement for *Akt1* in males during spatial memory formation.

## *Akt* deficiency impacts conditioned fear in an isoform- and sex-specific manner

To identify further isoform-specific contributions of AKT and sex-specific effects on cognitive processes, we examined associative fear long-term memory (LTM) in *Akt* mutant mice conditioned to associate an aversive foot-shock to an initially neutral tone. We found no effects on training performance or contextual and cued LTM evaluated 24 hr post-training in *Akt1* KO mice (*Figure 3A,B*). For *Akt2* KO mice, contextual LTM was impaired relative to WT littermates in males (*Figure 3C*; t (28)=-2.474, p=0.020) but not females (*Figure 3D*). In *Akt3* KO mice, we detected no fear conditioning effects (*Figure 3E,F*).

Because *AKT* mutations and single nucleotide polymorphisms (SNP)s have been linked to psychiatric disorders like schizophrenia (*Emamian et al., 2004*; *Ikeda et al., 2004*; *Schwab et al., 2005*; *Xu et al., 2007*) and extinction is altered in the disorder (*Craske et al., 2018*), we also examined extinction learning in *Akt* isoform mutants after fear conditioning. Interestingly, *Akt1* KO mice

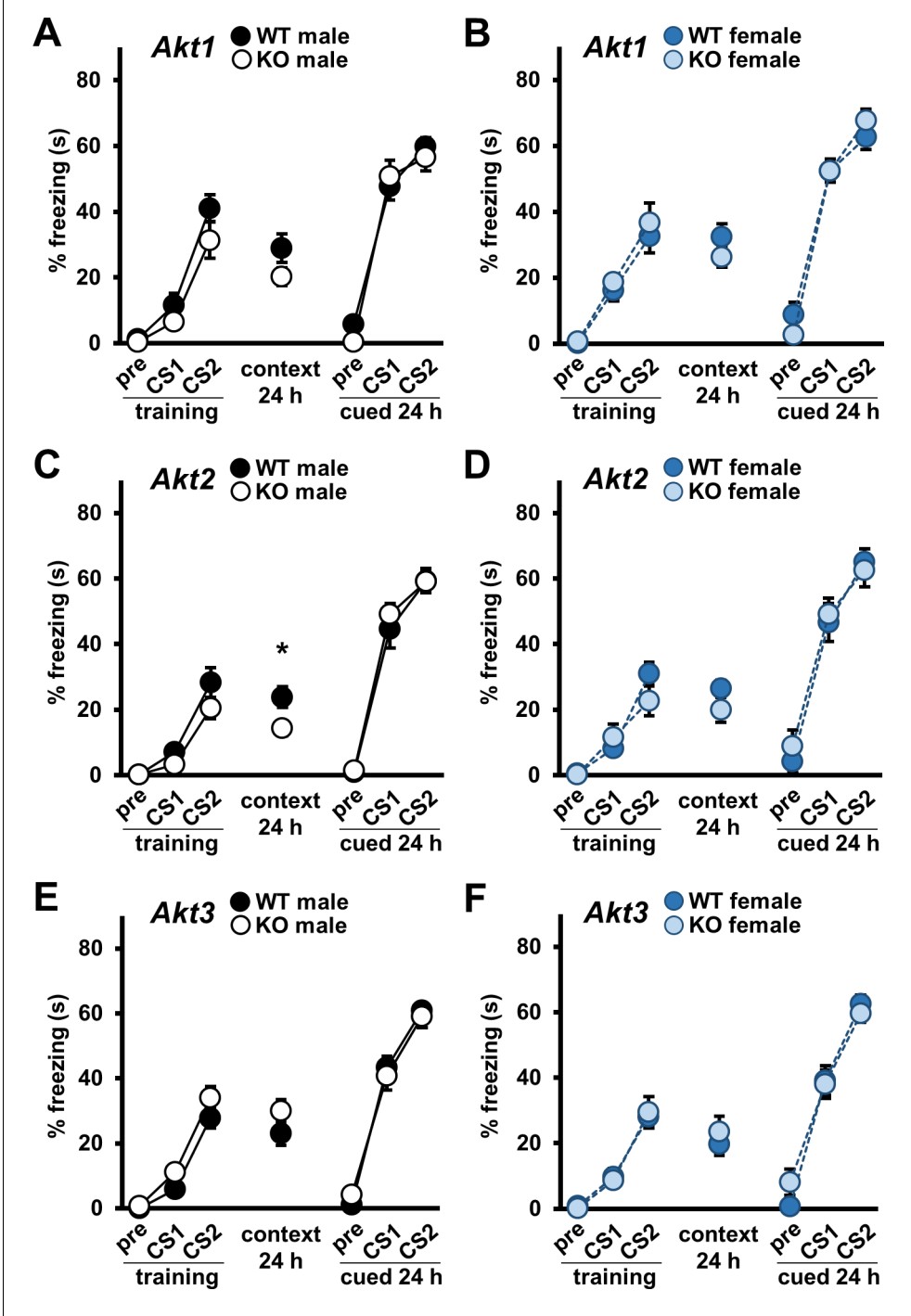

**Figure 3.** Single *Akt* isoform deficiency does not impact cued fear LTM but *Akt2* deficiency affects contextual fear LTM in males. Associative fear conditioning acquisition and long-term memory (LTM) in *Akt* isoform mutant male and female mice shown as percent time spent freezing. In training, freezing behavior was assessed during the baseline (pre) and 30 s after the first (CS1) and second (CS2) CS-US presentations. Contextual and cued LTM were assessed 24–25 hr after training by measuring freezing behavior during the entire test for contextual LTM and during the baseline (pre) and CS presentations (CS1, CS2) in the cued LTM test. Performance of (**A**) *Akt1* KO male, (**B**) *Akt1* KO female, (**C**) *Akt2* KO male, (**D**) *Akt2* KO female, (**E**) *Akt3* KO male, and (**F**) *Akt3* KO female mice. *Akt2* KO male mice showed impaired contextual LTM compared with WT controls. No other significant differences were detected between *Akt* isoform KO and WT mice. *$p < 0.05$. N = *Akt1* (WT-M = 16–17, KO-M = 13–15, WT-F = 10,

*Figure 3 continued on next page*

*Figure 3 continued*

KO-F = 10); *Akt2* (WT-M = 14, KO-M = 14–16, WT-F = 10–11, KO-F = 10); *Akt3* (WT-M = 18, KO-M = 18–20, WT-F = 17, KO-F = 16–17).

The online version of this article includes the following source data for figure 3:

**Source data 1.** Figure 3 source data.

showed a male-specific enhancement in extinction learning compared with WT controls (*Figure 4A, B*; males F(1,30)=8.571, p=0.006). No extinction acquisition differences were observed in *Akt2* KO (*Figure 4C,D*) or *Akt3* KO (*Figure 4E,F*) mice. To determine the efficacy of extinction training in *Akt* mutant mice, we tested the resulting expression of cued fear memory. Consistent with their more rapid extinction learning, extinction LTM was enhanced specifically in *Akt1* KO males compared with WT controls (*Figure 4A*; F(1,30)=9.121, p=0.005). When we tested renewal of extinguished fear by re-exposing mice to the CS in the original training context, *Akt1* KO male mice also showed reduced renewal relative to WT controls (*Figure 4A*; F(1,30)=7.187, p=0.012). With *Akt2* KO mice, we detected no differences in extinction LTM from WT controls (*Figure 4C,D*), but KO males showed enhanced fear renewal (*Figure 4C*; F(1,26)=5.969, p=0.022). *Akt3* KO mice showed no differences from WT controls in extinction LTM or renewal (*Figure 4E,F*). These data support a role for *Akt1* and *Akt2* in associative fear memory processes of males.

## Restoring AKT1 expression in the PFC rescues fear extinction processes in *Akt1* KO males

Because extinction memory relies on the prefrontal cortex (PFC) for normal display (*Courtney et al., 1998*; *Fallgatter, 2001*; *Bussey et al., 2012*), we next tested if restoring AKT1 activity in the PFC might rescue the observed extinction effects in male *Akt1* KO mice (*Figure 4*). To do this, we injected the PFC of male WT and *Akt1* KO mice with AAV vectors to express either AKT1-GFP or Cre-GFP as a sham control (*Figure 5A*). We confirmed AAV expression in both the prelimbic (PL) and infralimbic (IL) regions of the PFC (*Figure 5B*). Furthermore, we confirmed that expression of AKT1 in the PFC of *Akt1* KO males (*Akt1* KO+AKT1$_{PFC}$) is restored in neurons (*Figure 5C*), both excitatory and inhibitory neurons (*Figure 5—figure supplement 1*). We found no difference in conditioned fear learning or contextual and cued 24 hr memory in *Akt1* KO+AKT1$_{PFC}$ mice compared to sham-treated WT and KO males (*Figure 5D*). However, we found a significant effect on extinction learning (*Figure 5E*; F(2,34)=5.400, p=0.009). In agreement with our previous experiment (*Figure 4*), *Akt1* KO-sham mice showed enhanced extinction learning compared with WT-sham mice (p=0.009). *Akt1* KO+AKT1$_{PFC}$ males showed a significant difference in extinction learning from *Akt1* KO-sham mice (p=0.047) but not from WT-sham mice (p=0.875), consistent with the idea that PFC AKT1 activity underlies the extinction effects we observed in *Akt1* KO mice. We also found a significant difference in extinction LTM between *Akt1* KO-sham and WT-sham groups (*Figure 5E*; F(2,34)=4.388, p=0.020, post-hoc comparison p=0.035) but no significant difference between WT-sham and *Akt1* KO+AKT1$_{PFC}$ groups (post-hoc comparison p=0.985), supporting the notion that PFC-expressed AKT1 rescued *Akt1* KO effects on extinction LTM. Finally, we examined extinction renewal following PFC AKT1 expression. We did not observe a significant difference between WT-sham and *Akt1* KO+AKT1$_{PFC}$ groups (*Figure 5E*; F(2,29)=1.018, p=0.374, post-hoc comparison p=0.605). Unexpectedly, we also did not observe a significant difference between WT-sham and *Akt1* KO-sham groups (post-hoc comparison p=0.375), as we did with WT and *Akt1* KO mice in the previous experiment (*Figure 4*). However, the combined data support that idea that AKT1-mediated PFC function underlies effects of *Akt1* deficiency on the expression of extinction-related behaviors.

## Restricted *Akt1* deficiency affects contextual fear memory but not fear extinction

To determine if systemic or developmental *Akt1* deficiency led to the phenotypes we observed in *Akt1* KO males, we generated conditional KO mice with Cre-mediated *Akt1* removal (*Akt1* cKO) from excitatory neurons of the forebrain, including the amygdala and PFC (*Figure 6—figure supplement 1*) and hippocampus (*Levenga et al., 2017*), late in development (*Hoeffer et al., 2008*). We found no differences in OFA center time (*Figure 6A*) or EPM open arm time (*Figure 6B*) between

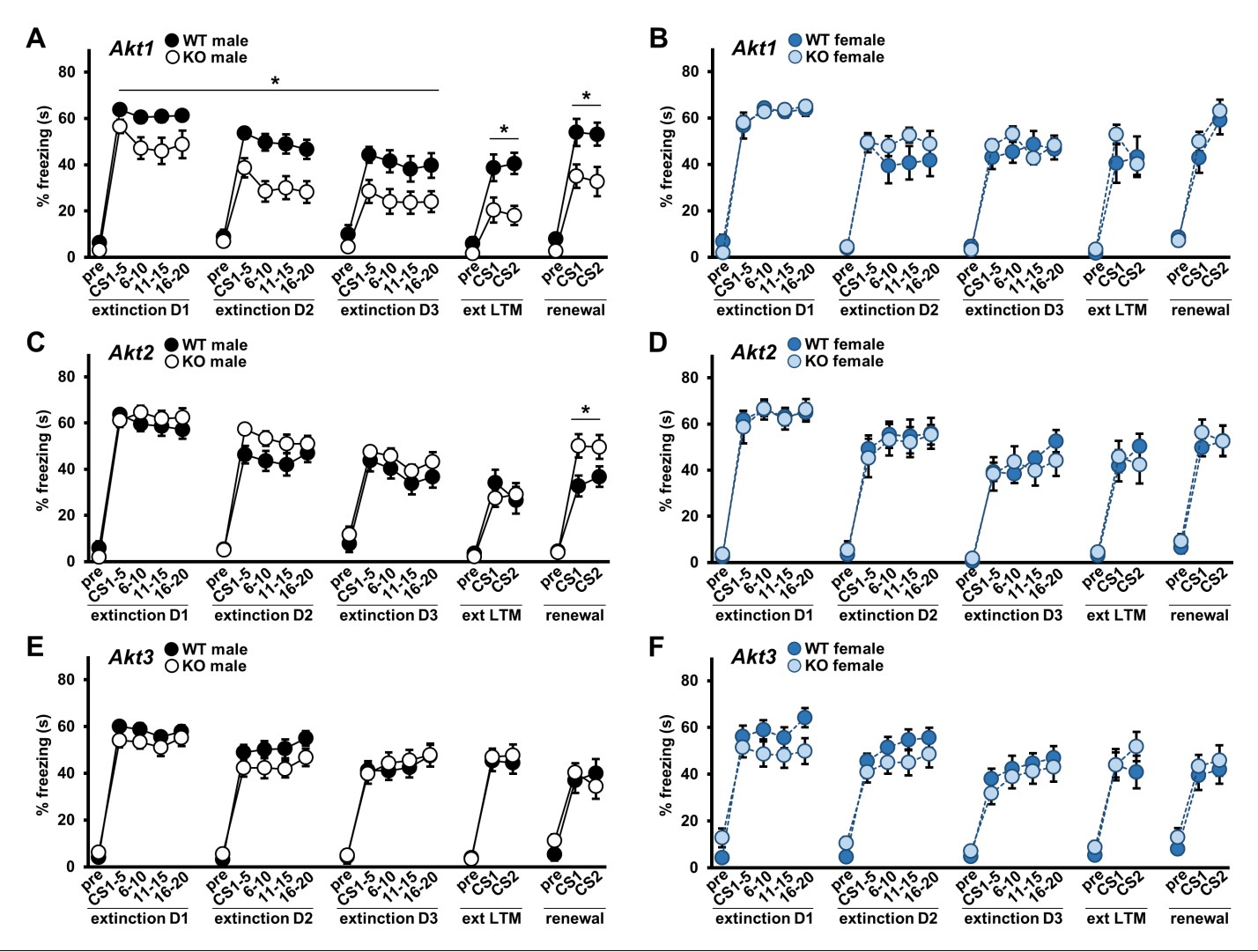

**Figure 4.** AKT1 deficiency enhances fear extinction learning and AKT2 removal enhances fear renewal. Extinction training to the CS was performed over three days (D1–D3) with 20 CS presentations each day. Extinction learning in mice measured as percent freezing during the baseline period prior to the first CS (pre) and during each CS presentation shown as average percent freezing in 4 blocks of 5 CS presentations each for every day of training. Efficacy of extinction learning assessed by testing extinction LTM (ext LTM) and renewal of extinguished cued LTM. Ext LTM measured as percent freezing to 2 CS presentations in a novel context 24 hr after completion of extinction training. Renewal measured as percent freezing to 2 CS presentations in the original training context 10 d after ext LTM testing. Performance of (A) *Akt1* KO male, (B) *Akt1* KO female, (C) *Akt2* KO male, (D) *Akt2* KO female, (E) *Akt3* KO male, and (F) *Akt3* KO female. *Akt1* KO males showed faster rates of extinction compared with controls while *Akt2* KO males showed increased renewal freezing compared with controls. No other significant differences were detected between *Akt* isoform KO and WT mice. *p<0.05. N = *Akt1* (WT-M = 17, KO-M = 15, WT-F = 10, KO-F = 10); *Akt2* (WT-M = 13–14, KO-M = 15–16, WT-F = 9, KO-F = 9); *Akt3* (WT-M = 18, KO-M = 20, WT-F = 15–17, KO-F = 15–16).

The online version of this article includes the following source data for figure 4:

**Source data 1.** Figure 4 source data.

male *Akt1* cKO and WT mice, suggesting AKT1 activity in forebrain excitatory neurons or late development is not required for normal anxiogenic-like responses. Additionally, *Akt1* cKO males performed similarly in the MWM to WT controls (**Figure 6C**), suggesting AKT1 activity in forebrain excitatory neurons or late development is not required for spatial memory as well. Conditional *Akt1* removal also had no effect on associative fear learning and cued LTM (**Figure 6D**) or on extinction learning and LTM (**Figure 6E**), although there was a trend for enhanced fear renewal (**Figure 6E**; F (1, 22)=3.105, p=0.092). Interestingly, contextual LTM was impaired in *Akt1* cKO males (**Figure 6D**; t (22)=2.535, p=0.019), suggesting AKT1 activity in forebrain excitatory neurons or late development

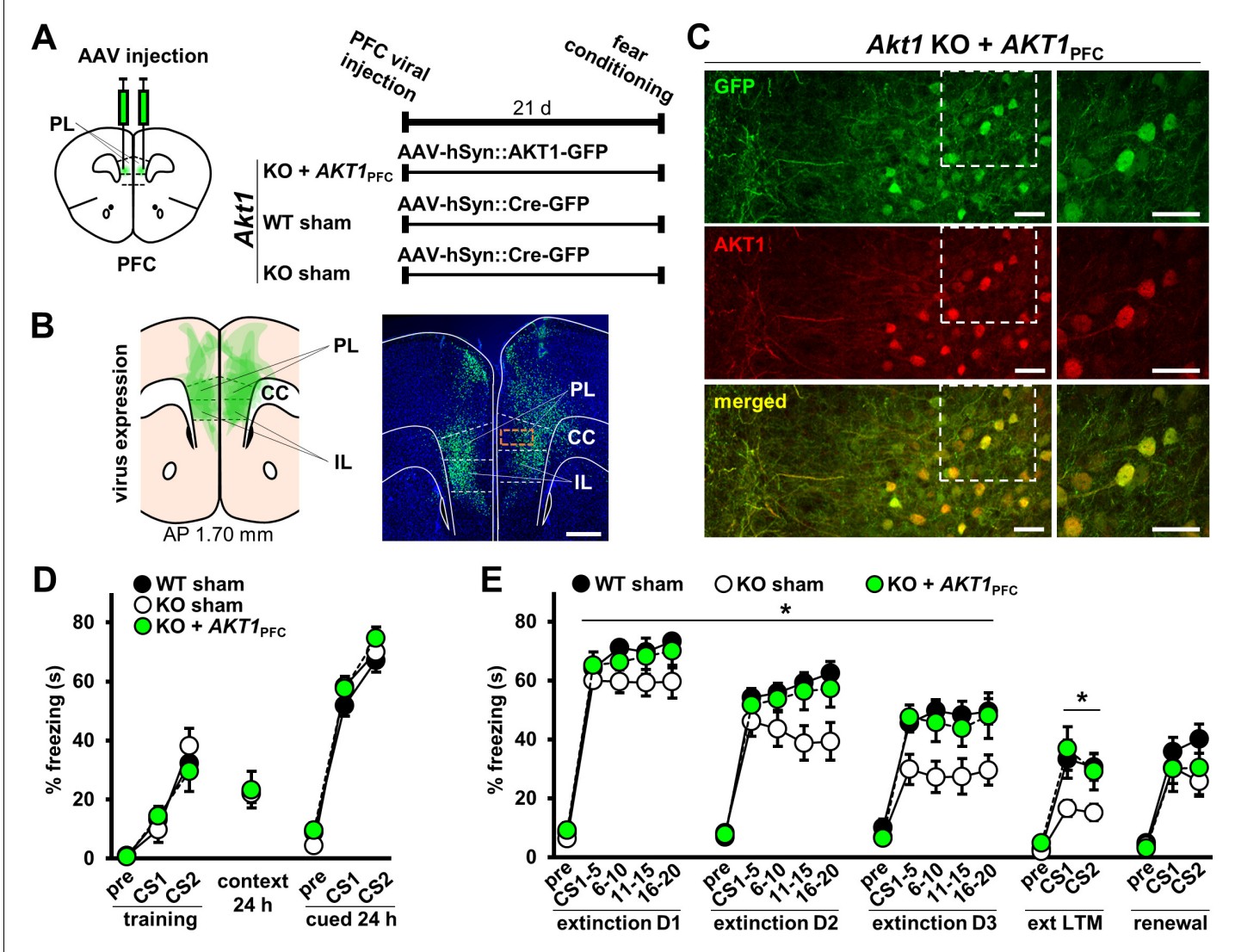

**Figure 5.** Restoration of AKT1 expression in the PFC normalizes extinction learning and LTM in *Akt1* KO males. (A) Experimental design to test the function of AKT1 in the PFC. AKT1 expression was restored using AAV-mediated pan-neuronal (hSyn) co-expression of AKT1 and green fluorescent protein (GFP) in the PFC of *Akt1* KO male mice (KO+$AKT1_{PFC}$). Sham surgeries were performed on *Akt1* KO and WT males using hSyn-driven expression of GFP-tagged Cre recombinase (Cre-GFP) to generate control groups (WT-sham, KO-sham). Mice were injected with AAV in the PL at 1.98 mm AP and then tested on associative fear conditioning and extinction beginning 21 days post-injection. (B) *Left:* Schematic of AAV expression spread (green areas) in the mouse PFC at 1.70 mm AP overlaid for six subjects (N = 2/group). *Right*: Representative image of AAV expression. Green, GFP; blue, Hoechst. Scale bar, 500 μm. (C) Higher magnification of PFC corresponding to orange boxed area in (B) right image. Staining for AKT1 (red) in the PFC of KO+$AKT1_{PFC}$ mice confirms that AKT1 expression is restored in neurons and colocalizes with GFP (AAV-infected cells). Scale bars, 40 μm. (D) Sham surgery and AKT1 restoration in *Akt1* KO mice did not affect fear acquisition or 24 hr contextual and cued LTM. (E) Extinction learning and LTM were restored to WT levels in KO+$AKT1_{PFC}$ mice but not renewal. KO-sham mice displayed reduced freezing compared with either WT-sham or KO+$AKT1_{PFC}$ groups. *p<0.05. N(WT-sham, KO-sham, KO+$AKT1_{PFC}$)=15,11,11 except for renewal, due to loss of animals between extinction and renewal test phases: N = 13,11,8. Schematics of the mouse brain are guided by *Franklin and Paxinos, 2007*. PFC, prefrontal cortex; PL, prelimbic cortex; IL, infralimbic cortex; CC, corpus callosum.

The online version of this article includes the following source data and figure supplement(s) for figure 5:

**Source data 1.** Figure 5 source data.
**Figure supplement 1.** Restoration of neuronal AKT1 expression in the PFC of *Akt1* KO male mice.

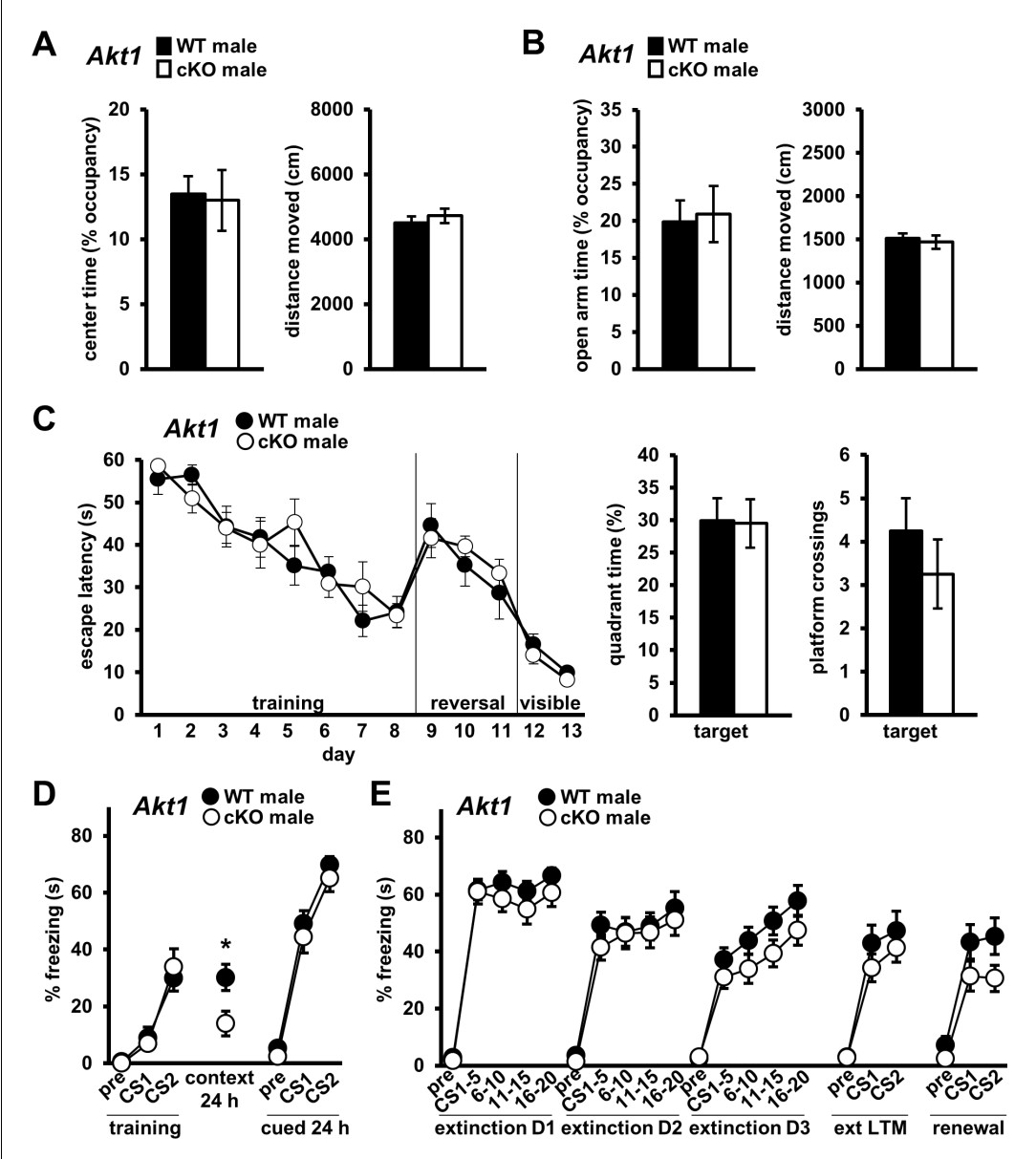

**Figure 6.** Conditional *Akt1* deficiency in excitatory neurons impairs contextual fear LTM but does affect anxiety-related behaviors or fear extinction processes. Behavior of conditional *Akt1* KO (*Akt1* cKO) male mice generated by removal of *Akt1* with a forebrain excitatory neuron-specific Cre driver in floxed *Akt1* mice. (A) OFA activity, measured as percent time spent exploring the center zone, and (B) EPM activity, measured as percent time spent in the open arms, were similar between *Akt1* cKO and WT mice. Distance moved during OFA or EPM assays also were similar between genotypes. N = 14 WT, 20 cKO. (C) *Akt1* cKO mice and WT controls showed similar latencies to escape the MWM during training, reversal, and visible platform phases and similar probe test performance measured as percent time spent and number of platform location crossings in the target quadrant. N = 8 WT, 8 cKO. (D) *Akt1* cKO mice showed impaired contextual fear LTM but no difference in fear learning or cued LTM compared with WT controls. N = 11 WT, 13 cKO. (E) No significant difference in extinction learning, extinction LTM (ext LTM) or renewal was seen between *Akt1* cKO and WT mice. N = 11 WT, 13 cKO. *$p < 0.05$.

The online version of this article includes the following source data and figure supplement(s) for figure 6:

**Source data 1.** Figure 6 source data.

**Figure supplement 1.** AKT1 expression in conditional *Akt1* KO (*Akt1* cKO) male mice under the excitatory neuron-specific T29-1 *Camk2α*-Cre driver.

of male mice is required for contextual fear memory. Although we did observe some effect on memory formation, remarkably we did not identify a requirement for excitatory neuron AKT1 activity in the normal expression of many behaviors. Taken together with our results from restoring neuronal AKT1 expression in the PFC (*Figure 5*), these data suggest that AKT1 activity within interneurons is required for normal extinction learning.

## AKT1 and AKT3 function may substitute for each other during memory processes

Because AKT1 and AKT3 are expressed in similar cell types in the brain (*Levenga et al., 2017*), they may provide compensatory activity in the absence of the other. To test this idea, we generated mice with conditional *Akt1* removal in the *Akt3* KO background (*Akt1* cKO *Akt3* KO) (*Levenga et al., 2017*). When we assessed anxiety-like behavior in *Akt1* cKO *Akt3* KO mice, we found no performance differences in the OFA (*Figure 7A*) or EPM (*Figure 7B*) compared to WT controls for either sex. In the MWM, spatial learning was also indistinguishable between *Akt1* cKO *Akt3* KO and WT mice for both sexes (*Figure 7C,D*). However, *Akt1* cKO *Akt3* KO mice did show significantly reduced target quadrant time and target platform crossings compared with WT littermates for both males (*Figure 7C*; quadrant time t(20)=2.172, p=0.042; crossings t(20)=3.406, p=0.003) and females (*Figure 7D*; quadrant time t(22)=3.262, p=0.004; crossings t(22)=2.283, p=0.032), indicating pronounced spatial memory deficits with simultaneous *Akt1* and *Akt3* removal. Similarly in associative fear conditioning, training performance of *Akt1* cKO *Akt3* KO mice was normal but contextual LTM

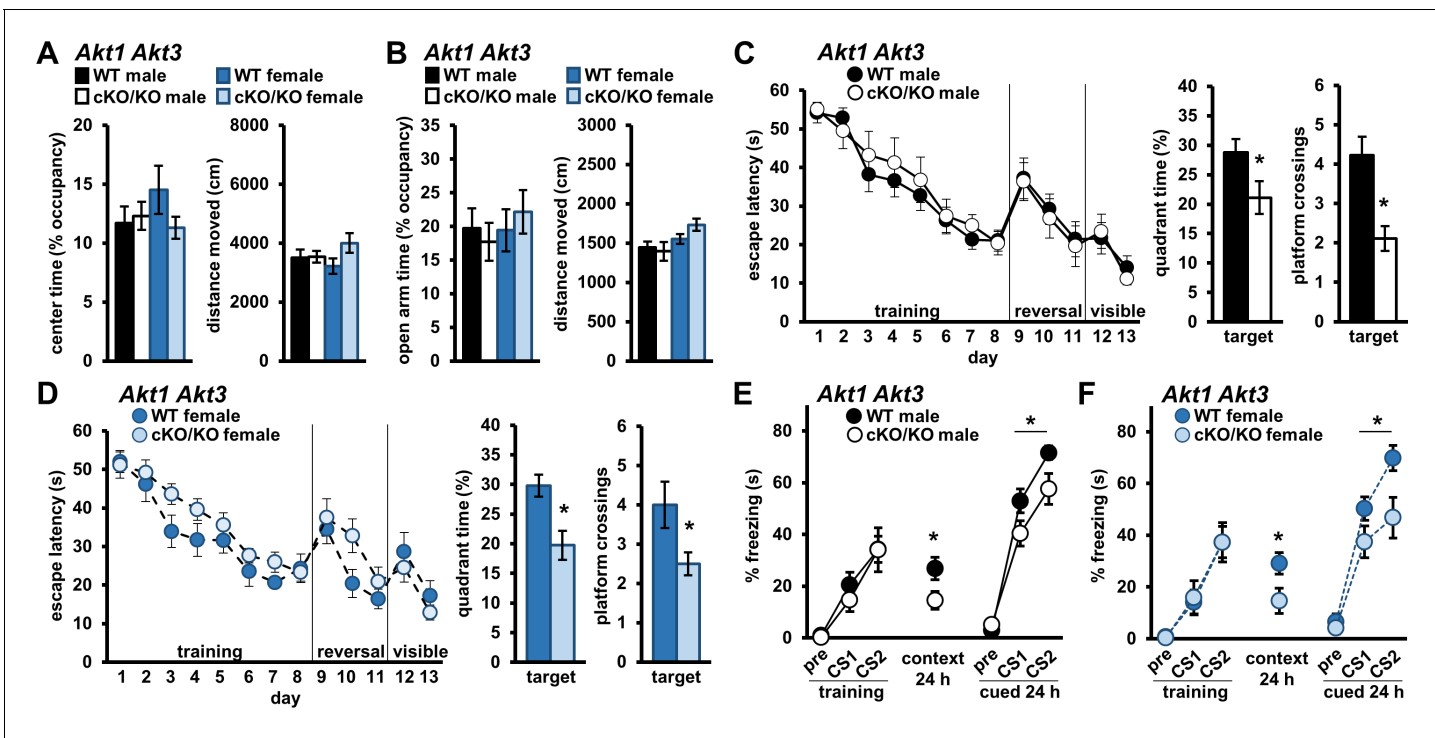

**Figure 7.** Simultaneous removal of AKT1 and AKT3 isoforms from the brain promotes memory deficits in both male and female mice. Male and female mice with conditional *Akt1* removal in the background of *Akt3* deficiency (*Akt1* cKO *Akt3* KO) were assessed in anxiety-related behavior, spatial memory, and associative fear memory tests. No differences were detected between *Akt1* cKO *Akt3* KO mutant mice and WT controls for both sexes in (**A**) OFA activity, measured as percent time spent exploring the center zone, and (**B**) EPM activity, measured as percent time spent in the open arms. Distance moved during OFA or EPM assays were similar between genotypes for both sexes. N = 15 WT-M, 13 cKO/KO-M, 16 WT-F, 15 cKO/KO-F. (**C**) Male and (**D**) female *Akt1* cKO *Akt3* KO mice had similar latencies to escape the MWM during training, reversal, and visible platform phases compared to their respective WT controls but showed significantly reduced target quadrant time and platform crossings. N = 13 WT-M, 9 cKO/KO-M, 12 WT-F, 12 cKO/KO-F. (**E**) Male and (**F**) female *Akt1* cKO *Akt3* KO mice displayed normal fear learning but impaired contextual and cued LTM compared with their respective WT controls. N = 11 WT-M, 11 cKO/KO-M, 14 WT-F, 10–11 cKO/KO-F. *p<0.05.

The online version of this article includes the following source data for figure 7:

**Source data 1.** Figure 7 source data.

was impaired for both males (*Figure 7E*; t(20)=2.213, p=0.039) and females (*Figure 7F*; t(23)=2.250, p=0.034) compared with WT controls. Taken together with the results in single *Akt* isoform KO mice (*Figures 2–3*), these data suggest that AKT3 substitutes for AKT1 activity in female mice while AKT1 can substitute for AKT3 in both sexes during spatial memory tasks. Furthermore, these findings suggest all three AKT isoforms play a role in contextual fear memory of male mice. Interestingly, *Akt1* cKO *Akt3* KO males and females also showed deficits in cued LTM (*Figure 7E,F*; males F(1,20) =5.176, p=0.034; females F(1,22)=4.533, p=0.045), suggesting *Akt1* and *Akt3* can compensate for each other in cued fear memory in both sexes. These data strongly implicate AKT1 and AKT3 in providing overlapping function in the brain. Importantly, they suggest that female behavior and memory are in fact regulated by AKT1 activity but AKT3 can provide compensatory function in females not available in male mice.

## AKT expression under normal and deficiency conditions are regulated similarly between sexes

Our data support the idea that AKT isoforms play differential roles in behavior, and importantly, these requirements are expressed in a sex-specific fashion. Behavioral results are summarized in *Table 1*. To determine if the basis for these sex differences were due to inherent variation between sexes or a sex-specific compensatory response to *Akt* isoform removal, we conducted biochemical analyses of AKT isoform expression between the sexes. To do this, we immunoblotted proteins from the brains of *Akt* isoform mutant mice and WT controls and measured AKT isoform and activation levels. Comparing WT males and females, we identified higher AKT2 levels in the hippocampus of female mice (*Figure 8A*; t(19)=3.045; p=0.007). No sex differences in protein levels were detected for the other isoforms or for total AKT levels measured with a pan-AKT antibody that recognizes all AKT isoforms (*Figure 8A*). When AKT activation was measured between the sexes using isoform-specific and pan-AKT serine 473 phosphorylation (pAKT) antibodies, we also found no differences in AKT1, AKT2, and total AKT activation levels (*Figure 8A*). These data suggest sex differences in brain expression levels of AKT2 but not the fraction of phosphorylated AKT2 while the other isoforms show similar expression and activation between sexes.

We next measured AKT expression and activation in *Akt* isoform mutants. Using the pan-AKT antibody, we found reduced total AKT levels in all male *Akt* isoform-specific mutants compared with WT controls (*Figure 8B*; *Akt1* KO t(12)=3.427, p=0.005; *Akt2* KO t(12)=2.282, p=0.042; *Akt3* KO t(8) =11.79; p<0.0001), with the greatest reduction in *Akt3* KO mice. Interestingly, when we examined AKT activation with the pan-pAKT antibody, we found increased pAKT levels in male *Akt1* KO mice compared with WT controls (*Figure 8B*; t(12)=2.541, p=0.026) but no differences in *Akt2* KO or *Akt3* KO males (*Figure 8B*). When we performed the same analyses in female *Akt* isoform-specific mutants and WT controls, we observed decreased total AKT levels in a pattern like that found in males (*Figure 8C*; *Akt1* KO t(14)=3.507, p=0.004; *Akt2* KO t(10)=2.253, p=0.048; *Akt3* KO t(7) =11.77, p<0.0001). Similarly, we found increased pan-pAKT levels in female *Akt1* KO mice compared with WT controls (*Figure 8C*; t(14)=3.456, p=0.004) and no differences in *Akt2* KO or *Akt3* KO females (*Figure 8C*). These data provide evidence for similar levels of total AKT isoform expression and activation between sexes in response to removing one *Akt* isoform.

We next sought to determine how AKT isoforms were individually regulated in the *Akt* mutants. Using the isoform-specific pAKT antibodies to assay AKT isoform activation, we found increased pAKT2 levels in both male and female *Akt1* KO mice compared with WT controls (*Figure 8D*; males t(12)=2.645, p=0.021; females t(14)=2.160, p=0.049), contributing to the higher pan-pAKT levels observed in *Akt1* KO mice (*Figure 8B,C*). In *Akt2* KO mice, we found no difference in pAKT1 levels in either males or females (data not shown), but pAKT1 levels were significantly higher in both male and female *Akt3* KO mice compared with WT controls (*Figure 8E*; males t(8)=3.130, p=0.014; females t(15)=3.724, p=0.002). Because *Akt3* KO and WT mice showed similar pan-pAKT levels (*Figure 8B,C*), the increased pAKT1 levels may compensate for the absence of pAKT3 in *Akt3* KO mice. However, isoform-specific antibodies for AKT3 phosphorylation are not available, so we did not assess pAKT3 levels. These data demonstrate that male and female mice respond similarly to *Akt* isoform deficiency by activating the other AKT isoforms.

**Table 1.** Summary of behavioral impacts of *Akt* isoform deficiency.

| Behavioral assay | Genotype and/or viral expression manipulation | | | | | |
|---|---|---|---|---|---|---|
| | *Akt1* KO | *Akt2* KO | *Akt3* KO | *Akt1* KO + virally expressed PFC AKT1 (male only) | *Akt1* cKO (male only) | *Akt1* cKO *Akt3* KO |
| OFA | reduced center time in **males** | no effect | no effect | n/a | no effect | no effect |
| EPM | reduced open arm time in **males** | reduced open arm time in **males** | no effect | n/a | no effect | no effect |
| MWM | reduced platform crossings during probe test in **males** | no effect | no effect | n/a | no effect | reduced quadrant time and platform crossings during probe test in **males** and **females** |
| Contextual fear LTM | no effect | impaired in **males** | no effect | no effect | impaired | impaired in **males** and **females** |
| Cued fear LTM | no effect | no effect | no effect | no effect | no effect | impaired in **males** and **females** |
| Fear extinction learning | enhanced | no effect | no effect | restored to WT levels | no effect | n/a |
| Fear extinction LTM | reduced in **males** | no effect | no effect | restored to WT levels | no effect | n/a |
| Fear extinction renewal | reduced in **males** | enhanced in **males** | no effect | no statistical difference from WT | no effect | n/a |

## Single *Akt* isoform deficiency results in minimal changes to some downstream and related AKT signaling pathways

To investigate the molecular mechanisms underlying the behavioral effects of *Akt* isoform deficiency that we observed, we performed a candidate pathway analysis of AKT signaling in the brain (*Figure 9—figure supplement 1*). AKT is known to be regulated upstream by protein 3-phosphoinositide-dependent protein kinase 1 (PDK1), a direct target of several neuronal surface signaling complexes (*Hoeffer and Klann, 2010*). PDK1 activity is regulated by phosphorylation of serine 241 (*Scheid et al., 2005*). A critical downstream substrate of AKT believed to play a role in neuropsychiatric disorders and be a target of psychotropic drugs used to treat the disorders is glycogen synthase kinase three beta (GSK3β) (*Beaulieu et al., 2004*; *Beaulieu et al., 2009*). AKT phosphorylates GSK3β at serine 9, downregulating its activity (*Cross et al., 1994*; *Cross et al., 1995*). The extracellular regulated kinase (ERK) pathway also has been implicated in neurological disorders and neural processes like extinction (*Radulovic and Tronson, 2010*; *Vithayathil et al., 2018*; *Albert-Gascó et al., 2020*) and is a parallel signaling pathway to AKT that can have crosstalk (*York et al., 2000*; *Perkinton et al., 2002*; *Sutton and Chandler, 2002*; *Jones et al., 2003*). ERK activity is regulated by phosphorylation of threonine 202/tyrosine 204 (ERK1) and threonine 185/tyrosine 187 (ERK2), which could be indirectly affected by AKT activity (*Ahn et al., 1990*; *Payne et al., 1991*).

Using western blotting, we examined brain tissue from *Akt* isoform mutant mice to determine if PDK1, GSK3β, and ERK1/2 signaling were affected in an isoform- or sex-specific fashion. Comparing *Akt1* KO and WT samples, we found no significant differences in phosphorylated or total levels of the proteins examined from the hippocampus for either sex (*Figure 9A,B*; *Supplementary file 1*). Because AKT1 loss affected extinction (*Figure 4*), we also examined signaling in the PFC of *Akt1* KO mice but found no differences in males (*Figure 9C*; *Supplementary file 1*). *Akt1* KO females, however, showed reduced pGSK3β levels in the PFC compared with WT controls (*Figure 9D*; t(12) =3.567; p=0.004). Comparing *Akt2* KO and WT hippocampal samples, we found no significant differences in the proteins examined for either sex (*Figure 9E,F*; *Supplementary file 1*). In the hippocampus of *Akt3* KO mice, we previously found reduced pGSK3β levels in males compared with WT controls (*Levenga et al., 2017*). In females, we found no difference in pGSK3β levels between *Akt3*

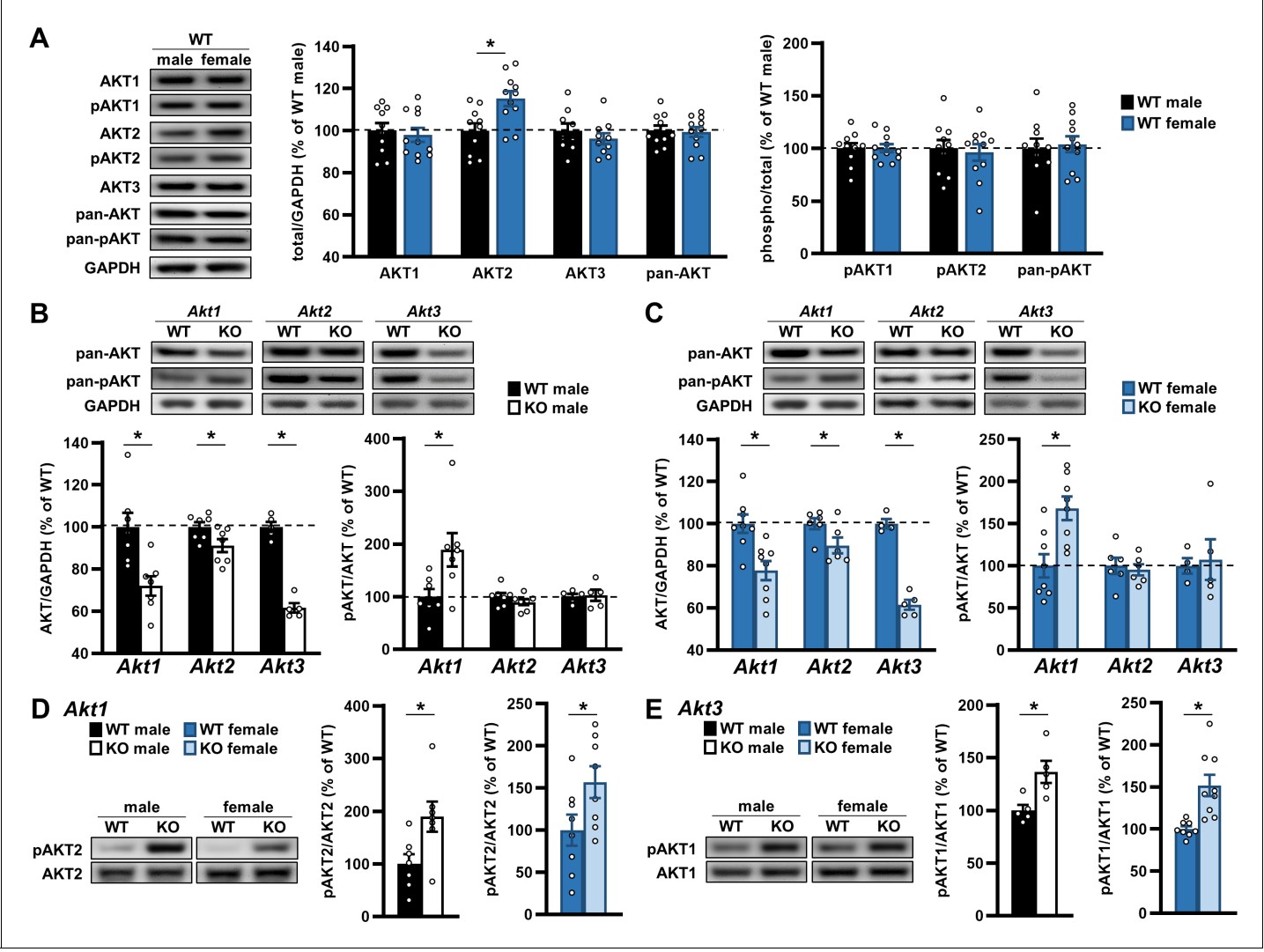

**Figure 8.** AKT expression under normal and deficiency conditions are regulated similarly between sexes. (A) Western blot analysis of AKT isoforms individually, detected with isoform-specific AKT antibodies, and all together, detected with pan-AKT antibodies, in the hippocampus of male versus female WT mice. *Left*: Representative blots. *Middle:* Total levels of each AKT isoform and pan-AKT normalized by GAPDH levels and shown as percent of immunoreactivity in WT males. *Right*: Phosphorylated (p) levels of individual and all (pan) AKT isoforms normalized by their respective total levels and shown as percent of immunoreactivity in WT males. No differences between the sexes were detected except higher AKT2 levels in female WT mice compared with males. N = AKT1,2, pan-AKT: 10 WT-M, 11 WT-F; AKT3: 9 WT-M, 10 WT-F. (B–C) Western blot analysis of total (pan-AKT) and phosphorylated (pan-pAKT) AKT levels in the hippocampus of (B) male and (C) female mice with single-isoform deletions of *Akt1, Akt2,* or *Akt3* compared with their respective WT controls. *Top*: Representative blots. *Left*: Total AKT levels normalized by GAPDH levels and shown as percent of immunoreactivity in WT controls. *Right*: Pan-pAKT levels normalized by pan-AKT levels and shown as percent of immunoreactivity in WT controls. Both sexes showed reduced total AKT levels with single *Akt* isoform deletions and an upregulation of pAKT levels with *Akt1* deficiency. N = *Akt1*: 7 WT-M, 7 KO-M, 8 WT-F, 8 KO-F; *Akt2*: 7 WT-M, 7 KO-M, 6 WT-F, 6 KO-F; *Akt3*: 5 WT-M, 5 KO-M, 4 WT-F, 5 KO-F. (D) Western blot analysis of pAKT2 levels normalized by total AKT2 levels in the hippocampus of *Akt1* KO male (*left graph*) and female (*right graph*) mice compared with their respective WT controls and shown as percent of immunoreactivity in WT controls. *Left*: Representative blots. Both sexes showed increased pAKT2 levels in response to *Akt1* deficiency. N = 7 WT-M, 7 KO-M, 8 WT-F, 8 KO-F. (E) Western blot analysis of pAKT1 levels normalized by total AKT1 levels in the hippocampus of *Akt3* KO male (*left graph*) and female (*right graph*) mice compared with their respective WT controls and shown as percent of immunoreactivity in WT controls. *Left*: Representative blots. Both sexes showed increased pAKT1 levels in response to *Akt3* deficiency. N = 5 WT-M, 5 KO-M, 8 WT-F, 9 KO-F. *p<0.05.

The online version of this article includes the following source data for figure 8:

**Source data 1.** Figure 8 source data.

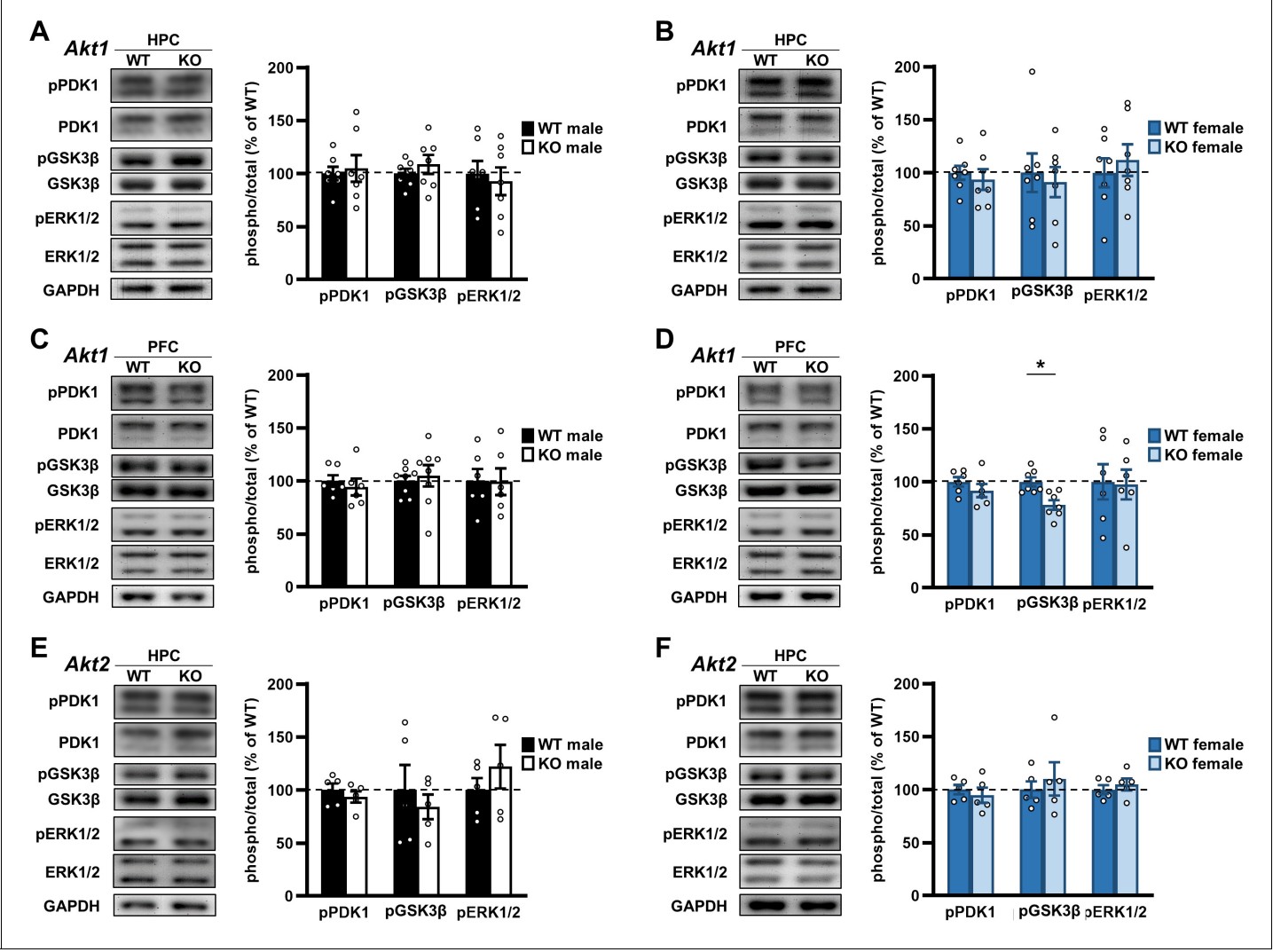

**Figure 9.** Examination of *Akt* isoform deficiency effects on neuromolecular signaling pathways. Upstream signaling of AKT was assessed by measuring phosphorylation levels of phosphoinositide-dependent protein kinase 1 (PDK1) at serine 241. Downstream signaling of AKT was assessed by measuring phosphorylation levels of glycogen synthase kinase three beta (GSK3β) at serine 9. Crosstalk between AKT and the extracellular regulated kinase (ERK) pathway was assessed by measuring ERK1 and ERK2 phosphorylation levels at threonine 204/tyrosine 202 and threonine 185/tyrosine 187, respectively. See *Figure 9—figure supplement 1* for schematic of pathways. (**A**) Western blot analysis of phosphorylated (p) proteins normalized by their respective total levels in the hippocampus (HPC) of *Akt1* KO males and shown as percent of immunoreactivity in WT controls. *Left*: Representative blots. N = 7 WT, 7 KO. (**B**) Western blot analysis of signaling in the HPC of *Akt1* KO females. N = 7 WT, 7 KO. (**C**) Western blot analysis of signaling in the prefrontal cortex (PFC) of *Akt1* KO males. pPDK1, pERK1/2: N = 6 WT, 6 KO; pGSK3β N = 8 WT, 8 KO. (**D**) Western blot analysis of signaling in the PFC of *Akt1* KO females showed reduced pGSK3β levels compared with WT controls. pPDK1, pERK1/2: N = 6 WT, 6 KO; pGSK3β N = 7 WT, 7 KO. (**E**) Western blot analyses of signaling in the HPC of *Akt2* KO males and (**F**) females compared to their respective WT controls. N = 5 WT-M, 5 KO-M, 5 WT-F, 5 KO-F. *p<0.05.

The online version of this article includes the following source data and figure supplement(s) for figure 9:

**Source data 1.** Figure 9 source data.
**Figure supplement 1.** Schematic of AKT-dependent and interacting signaling pathways.
**Figure supplement 2.** *Akt3* deficiency does not alter GSK3β activation in the hippocampus of female mice.
**Figure supplement 3.** Interneuron expression in the PFC of male and female WT mice.

KO and WT hippocampal samples (*Figure 9—figure supplement 2*). We also examined if interneuronal expression was different between the sexes. Because we posit that interneuronal AKT1 activity in the PFC is required for extinction learning in males based on our results (*Figures 4–6*), any sex difference in interneuronal expression in the PFC could contribute to the observed results. Using the

interneuronal markers parvalbumin (PV) and glutamate decarboxylase 67 (GAD67), we assessed the number of PV-positive interneurons and the level of GAD67 in the WT mouse PFC from both sexes and found no differences between males and females (*Figure 9—figure supplement 3*). While certainly not a comprehensive examination of *Akt* isoform deficiency impacts on signaling in the brain, these results may help to explain the isoform- and sex-specific behavioral effects we observed in *Akt* mutant mice and suggest additional pathways are involved.

## Discussion

We provide evidence for *Akt* isoform-specific functions underlying cognitive processes for affective behavior, spatial and contextual memory formation, and extinction learning and memory. Our study is the first to comprehensively investigate the behavioral consequences of *Akt* deficiency in the context of sex-specific effects, brain-specific *Akt1* deletion, and simultaneous *Akt1* and *Akt3* removal from the brain. We find that AKT1 loss more broadly affects neural function, impacting anxiety-related behavior, spatial memory, extinction learning, and renewal in *Akt1* KO mice but only the males. This may identify a role for *Akt1* outside the brain or during development that preferentially affects males. We also demonstrate that *Akt2* is required for the display of normal anxiety-like behavior and contextual fear memory. Although we found no evidence for *Akt3*-specific roles in the assays we performed, *Akt3* removal did enhance phenotypes in the *Akt1* mutant background, importantly extending them to females. All behavior results are summarized in *Table 1*. Finally, we found that the loss of individual AKT isoforms in the brain resulted in increased activation of the other isoforms and minimal changes to upstream PDK1, downstream GSK3β, and crosstalk with ERK signaling, with some sex-specific differences in AKT regulation and signaling pathways. Sex differences in neuropsychiatric disorders are well-recognized but poorly understood and understudied. By identifying sex-specific effects on AKT isoform-specific and redundant activity in cognitive processes, our study provides valuable new insight into how the highly influential AKT signaling pathway may affect diagnosis and treatment of neuropsychiatric disorders.

### AKT isoforms and behaviors linked to psychiatric disorders

Examining affective behavior, spatial and contextual memory, and extinction in mice has translational relevance to neuropsychiatric disorders. Anxiety is featured in many psychiatric conditions, including MDD, bipolar disorder (BD), and schizophrenia (*American Psychiatric Association, 2013*). Hippocampus-dependent memory functions such as spatial and episodic memory are impaired in schizophrenia (*Achim and Lepage, 2003*; *Titone et al., 2004*; *Achim and Lepage, 2005*; *Ragland et al., 2015*). Defects in extinction processes are observed in anxiety, depression, and schizophrenia (*Elliott et al., 1997*; *Boks et al., 2000*; *Maren and Holmes, 2016*; *Craske et al., 2018*). Extinction is also used in cognitive behavioral therapy to treat fear-associated disorders like PTSD and panic disorders (*Powers et al., 2017*). Therefore, delineating AKT isoform-relevant information about the neural circuitry and molecular signaling underlying these processes modeled in mice may aid therapeutic strategies.

Our study is the first to report on the effects of *Akt* deficiency in several murine behaviors and expands previous knowledge on the contribution of AKT isoforms. For affective behaviors, no prior study had examined *Akt1* KO mice. An earlier study reported higher anxiogenic-like behaviors in *Akt2* KO mice, as we report, but did not examine sex-specific effects (*Leibrock et al., 2013*). In *Akt3* KO mice, one earlier study reported increased anxiogenic-like EPM behavior (*Bergeron et al., 2017*) while another reported hyperactivity but no anxiety-like behavioral alterations (*Howell et al., 2017*). The discrepancies may result from using smaller sample sizes than our study, which can cause cohort effects, and mixing sexes. For spatial memory, previous studies had tested *Akt1* KO mice in the MWM but did not examine sex-based performance comprehensively and reported conflicting results (*Balu et al., 2012*; *Chang et al., 2016*; *Wang et al., 2017*). While our results support *Balu et al., 2012*, reasons for discrepancies with the other studies are unclear, but in *Chang et al., 2016* the unusually superior MWM performance of WT females, achieving maximal training performance after just one day, makes the apparently impaired KO performance difficult to interpret. Only one prior study examined MWM performance in *Akt2* KO mice and found no effects (*Leibrock et al., 2013*), confirmed by our results for both sexes. In *Akt3* KO mice, previous MWM studies had been conducted with two reporting similar findings to ours (*Bergeron et al., 2017*; *Howell et al., 2017*) and

another reporting a mild learning impairment but normal memory (*Wang et al., 2017*). For associative fear conditioning, unlike our results, a previous study reported impaired contextual fear memory in *Akt1* mutants (*Balu et al., 2012*) but used far fewer mice than our study and did not consider sex-specific effects. Fear extinction in *Akt1* mutants and both fear conditioning and extinction in *Akt2* mutants had not been studied previously. A prior study examined fear extinction in *Akt3* KO mice using a different protocol but found no effects (*Howell et al., 2017*), in agreement with our findings. Finally, no prior study had examined the behavioral impacts of conditional AKT1 deficiency or loss of two isoforms simultaneously. Our results suggest that AKT1 and AKT3 have overlapping functions but exert their effects in a sex-specific fashion, perhaps pointing to compensatory activities present in females but not males, whereas AKT2 has unique but also sex-specific functions.

These results are important given our previous work and others showing differential AKT isoform expression in the brain (*Zeisel et al., 2015*; *Levenga et al., 2017*; *DuBois et al., 2019*). AKT1 and AKT3 are expressed in overlapping neuronal populations but only AKT1 is found in interneurons (*Levenga et al., 2017*) while only AKT3 is found in oligodendrocytes (*DuBois et al., 2019*). This may explain the synergistic effects of *Akt1* and *Akt3* removal on memory and why *Akt1* removal alone was sufficient to affect anxiety-like behavior and fear extinction: GABAergic mechanisms linked to interneuronal function were shown previously to mediate these behaviors (*Delamater et al., 2009*; *Maren et al., 2013*). Indeed, our rescue experiment with viral-mediated AKT1 restoration in the PL and IL regions of the PFC of *Akt1* KO mice highlight a critical role for AKT1 in these brain regions for extinction processes (*Figure 5*). This experiment contrasts with the results of our conditional *Akt1* KO studies, where *Akt1* removal from excitatory neurons of the forebrain (*Figure 6—figure supplement 1*) did not affect extinction (*Figure 6*). Our AKT1 restoration experiment does not distinguish between AKT1 activity in excitatory and inhibitory neurons for rescuing extinction phenotypes in *Akt1* KO mice (*Figure 4*); therefore, the results of our conditional *Akt1* removal data suggest that AKT1 activity within interneurons is required for normal extinction learning. Future efforts aimed at refining AKT1 activity using interneuronal restoration of AKT1 expression or interneuron-specific *Akt1* removal may help to resolve this question. Another implication of our viral rescue result taken together with our conditional *Akt1* removal experiments is that AKT1 activity is unlikely to be required during perinatal development to affect cognition and memory (*Figures 5* and *6*); AKT1 activity appears to be required during active neurobiological processes rather than developmentally. AKT3 may be able to compensate for AKT1 during development but not post-developmentally in active neurobiological processes like cognition and synaptic plasticity (*Figure 6* and *Levenga et al., 2017*). The behavioral effects of *Akt2* deficiency are also interesting because AKT2 is not expressed in neurons but rather astrocytes of the mouse brain (*Levenga et al., 2017*), and little has been known about astrocyte function in cognitive processes. Because initial learning is normal in the assays we performed (*Figures 2*, *3* and *6*), sensory processing is likely intact in the *Akt* mutants we studied, but our experiments do not distinguish between consolidation or retrieval processes. Future experiments using reversible temporal control of AKT activity may be used to address these questions.

## Human and animal studies linking AKT to psychiatric disorders

Abnormal AKT signaling has emerged as a potential mechanism underlying several psychiatric disorders. Schizophrenia, for example, has been associated with SNPs in *AKT1* (*Ikeda et al., 2004*; *Tan et al., 2008*; *Blasi et al., 2011*; *Karege et al., 2012*) and *AKT3* (*Psychosis Endophenotypes International Consortium et al., 2014*), reduced AKT1 levels in patient brains (*Emamian et al., 2004*), and pathogenic AKT signaling (*Liu et al., 2013*). Other studies have failed to find statistically robust links between schizophrenia candidate genes like *AKT* and the disorder (*Loh et al., 2013*; *Purcell et al., 2014*; *Farrell et al., 2015*; *Johnson et al., 2017*). However, phenotypic and genetic heterogeneity associated with complex brain disorders like schizophrenia, combined with divergent clinical diagnoses, may impede current genomic studies from capturing AKT contributions. Additionally, post-translational modification of AKT is critical for its activity and highly heritable (*Hutz et al., 2011*) yet seldom considered in genetic association studies. Functional studies provide strong evidence for AKT involvement in schizophrenia. Magnetic resonance imaging studies have linked AKT signaling and SNPs in the pathway to structural and functional brain abnormalities in schizophrenia patients (*Jagannathan et al., 2010*; *Nicodemus et al., 2010*; *Wolthusen et al., 2015*). Antipsychotics for treating schizophrenia, which antagonize dopamine receptor-2 function, activate AKT and

inhibit its substrate GSK3 (*Emamian et al., 2004*; *Alimohamad et al., 2005*; *Li et al., 2007*; *Beaulieu et al., 2009*). Furthermore, genetic and pharmacological manipulations of the AKT/GSK3 pathway affect dopaminergic signaling and schizophrenia-related behaviors (*Beaulieu et al., 2004*; *Emamian et al., 2004*; *Gould et al., 2007*). Atypical antipsychotics also promote AKT activity (*Alimohamad et al., 2005*; *Li et al., 2007*). In other disorders, AKT has been implicated by post-mortem brain tissue from suicide victims showing decreased AKT activity (*Hsiung et al., 2003*), genetic studies associating *AKT1* SNPs with BD (*Karege et al., 2010*; *Karege et al., 2012*) and depression (*Ellsworth et al., 2013*; *Pereira et al., 2014*), and pharmacological studies showing that lithium, used widely to treat schizophrenia and BP (*Chalecka-Franaszek and Chuang, 1999*; *De Sarno et al., 2002*; *Beaulieu et al., 2004*; *Beaulieu et al., 2008*; *Nciri et al., 2013*), and the anti-depressant ketamine (*Park et al., 2014*) induce AKT signaling. Altogether, these studies suggest AKT is a key factor in the manifestation of psychiatric disorders and a key mechanism of action by current treatments. Because these disorders are heterogeneous and available therapies still have limited effectiveness, our data identifying specific and redundant roles for AKT isoforms, combined with the cell-specific expression of AKT in the brain (*Levenga et al., 2017*; *DuBois et al., 2019*), will be useful for subdividing these disorders to improve diagnostic measures and develop more targeted treatments.

## Sex-specific effects of AKT signaling?

Sex-dependent differences in mental health are well-documented. Women are disproportionately impacted by affective disorders (*Rubinow and Schmidt, 2019*), with higher rates of MDD (*Kessler et al., 1993*; *Weissman et al., 1993*; *Kessler et al., 2005*) and anxiety disorders (*Yonkers et al., 2003*; *Maeng and Milad, 2015*). In men, schizophrenia is more prevalent (*Abel et al., 2010*) but over the course of normal aging, schizophrenia impacts women at higher rates (*Häfner, 2003*; *Meesters et al., 2012*). Sex-dependent effects also are evident in treatment profiles for neuropsychiatric disorders. Studies have found that women are more responsive to available treatments for schizophrenia and BD (*Seeman, 2012*; *Crawford and DeLisi, 2016*; *Alberich et al., 2019*). In MDD, women generally report better treatment effects but also worse outcomes during tricyclic treatment (*Keers and Aitchison, 2010*). In female mice, ketamine exerts more potent and rapid effects in the forced swim test compared with males (*Franceschelli et al., 2015*). Only one prior study has examined sex-dependent effects of AKT signaling in a neurological context, showing *Akt1* KO female but not male mice were resistant to pentylenetetrazol-induced epileptic effects compared with controls (*Chang et al., 2016*), which is consistent with the generally increased susceptibility of *Akt1* KO males to behavioral alterations in the present study. Sex-specific differences in *Akt* mutant behavior reported here and previously cannot be explained by AKT expression or activation differences between the sexes. Our data show that with the exception of AKT2 total levels, male and female mice exhibit similar AKT isoform levels (*Figure 8A*), activation under *Akt* deficiency conditions (*Figure 8B,C*), and compensatory responses to AKT isoform loss (*Figure 8D,E*). Because AKT levels and activation are largely similar between males and females, sex-related differences in behavior are likely defined by sex-related differences in AKT signaling and interacting pathways. To begin to address this idea, we examined candidate signaling upstream, downstream, and parallel of AKT (*Figure 9* and *Levenga et al., 2017*). Interestingly, we found reduced phosphorylation of the AKT substrate GSK3β in the PFC of *Akt1* KO females. Why this was not observed in males is of significant future interest. It may indicate that for some downstream AKT targets, such as GSK3β, males may recruit additional signaling to modulate their activities. These additional or compensatory pathways that normalize GSK3β signaling in males may contribute to the behavioral differences observed between male and female *Akt1* mutant mice (*Figures 1–4*). By contrast, pGSK3β levels were reduced in the hippocampus of *Akt3* KO male mice but unchanged in females compared with WT controls (*Figure 9—figure supplement 2* and *Levenga et al., 2017*). This may indicate that AKT substrates also are regulated in *Akt* isoform-specific pools, each with specialized regulation. These findings while very preliminary, do indicate that sex-specific differences in AKT-dependent or -interacting signaling pathways exist. Future investigation using more comprehensive and unbiased methodologies like RNAseq or protein mass spectrometry should help identify these sex-specific signaling differences. Our findings greatly extend understanding of sex-specific AKT isoform effects in cognitive processes, which further supports the need for more specialized diagnoses and

treatments. Future studies will be important to determine how AKT is differentially expressed and regulated in the brain between males and females.

## Conclusion

Neuropsychiatric disorders are major health and economic concerns, with many afflicted individuals going undiagnosed or misdiagnosed and inadequately treated. Given the links between AKT and neuropsychiatric disorders, more studies like ours are needed to increase understanding of how this potent molecular pathway, with multiple isoforms, impacts behaviors and neurobiological processes modeled for these disorders. Because treatment responses can differ based on sex, newer more effective treatments may be developed if sex-dependent differences in critical neuropsychiatric signaling pathways, such as AKT, are better understood.

# Materials and methods

**Key resources table**

| Reagent type (species) or resource | Designation | Source or reference | Identifiers | Additional information |
|---|---|---|---|---|
| Antibody | anti-AKT1 (Rabbit monoclonal) | Cell Signaling | Cat# 2938; RRID:AB_915788 | WB (1:2000) |
| Antibody | anti-AKT1 (Rabbit monoclonal) | Cell Signaling | Cat# 75692; RRID:AB_2716309 | IHC (1:100) |
| Antibody | anti-AKT1 phospho-S473 (Rabbit monoclonal) | Cell Signaling | Cat# 9081; RRID:AB_11178946 | WB (1:1000) |
| Antibody | anti-AKT2 (Rabbit monoclonal) | Cell Signaling | Cat# 2964; RRID:AB_331162 | WB (1:1000); IHC (1:100) |
| Antibody | anti-AKT2 phospho-S474 (Rabbit monoclonal) | Cell Signaling | Cat# 8599; RRID:AB_2630347 | (1:1000) |
| Antibody | anti-AKT3 (Mouse monoclonal) | Cell Signaling | Cat# 8018; RRID:AB_10859371 | WB (1:2000) |
| Antibody | anti-AKT3 (Rabbit monoclonal) | Cell Signaling | Cat# 14982; RRID:AB_2716311 | IHC (1:100) |
| Antibody | anti-pan-AKT (Rabbit monoclonal) | Cell Signaling | Cat# 4685; RRID:AB_2225340 | WB (1:5000) |
| Antibody | anti-pan-AKT phospho-S473 (Rabbit monoclonal) | Cell Signaling | Cat# 4058; RRID:AB_331168 | WB (1:2000) |
| Antibody | anti-GAPDH (Rabbit monoclonal) | Cell Signaling | Cat# 5174; RRID:AB_10622025 | WB (1:20000) |
| Antibody | anti-β-actin (Mouse monoclonal) | Cell Signaling | Cat# 3700; RRID:AB_2242334 | WB (1:20000) |
| Antibody | anti-PDK1 phospho-S241 (Rabbit polyclonal) | Cell Signaling | Cat# 3061; RRID:AB_2161919 | WB (1:5000) |
| Antibody | anti-PDK1 (Rabbit polyclonal) | Cell Signaling | Cat# 3062; RRID:AB_2236832 | WB (1:1000) |
| Antibody | anti-GSK3β phospho-S9 (Rabbit monoclonal) | Cell Signaling | Cat# 5558; RRID:AB_10013750 | WB (1:5000) |
| Antibody | anti-GSK3β (Rabbit monoclonal) | Cell Signaling | Cat# 9315; RRID:AB_490890 | WB (1:5000) |
| Antibody | anti-ERK1/2 phospho-T202/Y204 (Rabbit polyclonal) | Cell Signaling | Cat# 9101; RRID:AB_331646 | WB (1:2000) |
| Antibody | anti-ERK1/2 (Rabbit polyclonal) | Cell Signaling | Cat# 9102; RRID:AB_330744 | WB (1:7500) |
| Antibody | anti-NeuN (Mouse monoclonal) | Novus | Cat# NBP1-92693; RRID:AB_11036146 | IHC (1:1000) |

*Continued on next page*

*Continued*

| Reagent type (species) or resource | Designation | Source or reference | Identifiers | Additional information |
|---|---|---|---|---|
| Antibody | anti-GAD67 (Mouse monoclonal) | Millipore | Cat# MAB5406 RRID:AB_2278725 | IHC (1:1500) WB (1:10000) |
| Antibody | anti-paravalbumin (Mouse monoclonal) | Millipore | Cat# MAB1572; RRID:AB_2174013 | IHC (1:1000) |
| Antibody | anti-rabbit Cy3 (Donkey polyclonal) | Jackson ImmunoResearch | Cat# 711-165-152; RRID:AB_2307443 | (1:250) |
| Antibody | anti-mouse IgG2b Alexa 647 (Goat polyclonal) | Invitrogen | Cat# A-21242; RRID:AB_2535811 | (1:500) |
| Antibody | anti-mouse IgG1 Alexa 488 (Goat polyclonal) | Invitrogen | Cat# A-21121; RRID:AB_2535764 | (1:500) |
| Antibody | anti-mouse HRP (Goat polyclonal) | Promega | Cat# W4021; RRID:AB_430834 | (1:5000–20000) |
| Antibody | anti-rabbit HRP (Goat polyclonal) | Promega | Cat# W4011; RRID:AB_430833 | (1:5000–20000) |
| Other | Hoechst | Thermo Fisher Scientific | Cat# H3569; RRID:AB_2651133 | (1:3000) |
| Genetic reagent (*Mus musculus*), both sexes | *Akt1*[tm1Mbb], C57BL/6 (*Akt1* KO) | Jackson Laboratory | Stock # 004912; RRID:IMSR_JAX:004912 | |
| Genetic reagent (*Mus musculus*), both sexes | *Akt2*[tm1.1Mbb], C57BL/6 (*Akt2* KO) | Jackson Laboratory | Stock # 006966; RRID:IMSR_JAX:006966 | |
| Genetic reagent (*Mus musculus*), both sexes | *Akt3*[tm1.3Mbb], C57BL/6 (*Akt3* KO) | *Easton et al., 2005*; PMCID:PMC549378 | MGI Cat# 3804003, RRID:MGI:3804003 | Obtained from Birnbaum lab (UPenn) |
| Genetic reagent (*Mus musculus*), both sexes | *Akt1*[tm2.2Mbb], C57BL/6 (*Akt1*[fl/fl]) | Jackson Laboratory | Stock #026474; RRID:IMSR_JAX:026474 | |
| Genetic reagent (*Mus musculus*), females only | *Tg*[(CamkIIa-Cre)T29Stl], C57BL/6 (*Camk2a*-Cre) | *Hoeffer et al., 2008*; PMCID:PMC2630531 | MGI Cat# 6273652, RRID:MGI:6273652 | Obtained from Kelleher lab (MIT) before the Tonegawa lab submitted a strain to Jax with same name; **not** same strain as listed at JAX. Expression properties for this line in *Hoeffer et al., 2008*. |
| Recombinant DNA reagent | hSyn-hAKT1-hSyn-eGFP (AAV) | Vector Biolabs | RRID:SCR_011010 AKT1:Genbank RefSeq# BC000479.2 | Commercially produced custom construct |
| Recombinant DNA reagent | hSyn-eGFP-Cre (AAV) | Penn Vector Core | RRID:Addgene 105540 | |
| Software, algorithm | IBM SPSS Statistics | IBM Analytics | RRID:SCR_002865 | |
| Software, algorithm | Prism | GraphPad | RRID:SCR_002798 | |
| Software, algorithm | ImageQuant TL | GE Healthcare | RRID:SCR_014246 | |
| Software, algorithm | Icy | Institut Pasteur and France-BioImaging | RRID:SCR_010587 | Open source image processing |

## Mice

Single-isoform knockout (KO) mice for *Akt1*, *Akt2*, and *Akt3* as well as *Akt1*[fl/fl] *Akt3 KO* mice were generated on a C57B/l6 background as previously described (*Levenga et al., 2017*). To generate mice with conditional removal of *Akt1* in forebrain excitatory neurons, we bred *Akt1*[fl/fl] or *Akt1*[fl/fl] *Akt3 KO* mice with *Akt1*[fl/+] crossed to *Camk2a::Cre* mice on a C57Bl/6 background (*Hoeffer et al., 2008*). Because it would have been logistically prohibitive to generate all *Akt* mutant combinations from a single progenitor line, *Akt* mutant strains were maintained and tested separately. Mice from all strains were assessed for general health and sensory and locomotor capacities compared to their wild-type (WT) littermates. Apart from previously reported significant size differences in *Akt1* and

*Akt3* mutant backgrounds (*Cho et al., 2001*; *Easton et al., 2005*) and lower than expected Mendelian frequencies for *Akt1* KO mice (*Cho et al., 2001*), KO and WT mice were indistinguishable from each other. We also observed no significant differences in mortality during the testing period. Mice were group-housed in the same facility and maintained on a 12:12 hr light:dark schedule with food and water available *ad libitum*. Mice of both sexes were used in all experiments and tested over multiple independent cohorts at 3–6 months old. All procedures were approved by the University of Colorado, Boulder's Institutional Animal Care and Use Committee and conformed to the National Institutes of Health's *Guide for the Care and Use of Laboratory Animals*.

## Behavioral assays

Timeline: A schematic representation of the experimental timeline is shown in *Figure 1—figure supplement 1*. For *Figures 1–4* and *6*, mice were tested first in the OFA, then EPM the next day, followed by either MWM testing over 13 days or associative fear memory and extinction testing over 20 days. For *Figure 5*, mice were allowed to recover from surgery for 21 days and handled for three consecutive days, 30 min each day, immediately prior to fear conditioning. For *Figure 7*, mice were tested first in the OFA, then EPM the next day, followed by either MWM testing over 13 days or associative fear memory testing. Mice were acclimated to the testing room for 1 hr before each assay. All studies were performed with experimenters blind to genotype.

OFA: Mice were allowed 10 min to explore a white Plexiglas arena (40 × 40 cm$^2$) with 180 lux overhead lighting and 55 dB white noise present for their entire duration in the testing room. Data were collected and analyzed using the Ethovision XT video tracking system (Noldus, Wageningen, Netherlands), with the center zone defined as the area 10 cm from the arena walls.

EPM: Mice were allowed 5 min to explore a white EPM arena (30 cm arm length) under similar testing conditions to OFA and activity was analyzed with Ethovision XT as previously described (*Hoeffer et al., 2013*).

MWM: Mice were trained over 8 days as previously described (*Wong et al., 2015*) to locate a hidden escape platform 2–3 cm below the surface of a pool (112 cm diameter) of opaque water using visual cues outside of the pool. After the probe test on day 8, a reversal phase was introduced on days 9–11 in which mice were similarly trained to locate the hidden platform in the opposite quadrant. On days 12–13, mice were tested for visual acuity using a visible escape platform as previously described (*Wong et al., 2015*). Data were collected and analyzed using Ethovision XT.

Fear conditioning and extinction: Mice were trained on day one with two pairings of a tone (CS, 30 s, 85 dB white noise) and foot-shock (US, 2 s, 0.5-mA) presentation, and freezing behavior was measured to assess fear acquisition. On day 2, contextual and cued long-term memory (LTM) were assessed by re-exposing mice to the training context (white light, grid floor, and peppermint odor) or to the CS in a novel context, respectively, with no US. Order of testing contextual and cued LTM with 1 hr between tests for each mouse was randomly counterbalanced among animals. Memory was measured as percent time spent freezing during the 5 min contextual test or during the two 30 s CS presentations given 1 min apart in the cued test. On day 7, mice received extinction training in the cued LTM testing environment over three consecutive days with a 35 min session each day consisting of 20 (30 s) CS exposures at varying intervals. On day 10, extinction LTM was assessed like cued LTM in a novel context. On day 20, renewal of conditioned fear was performed by re-exposing mice to the CS in the original training context using the same protocol as for training except no US was delivered. Baseline freezing was also monitored prior to the first CS presentation in any session. Context novelty was generated using alternate lighting (red light), walls and flooring (changeable acrylic inserts with different display patterns or textures), and odorants (vanilla or lavender) compared to the training environment.

## Intra-PFC AAV injections

Akt1 WT and KO male mice 3–4 months old were anesthetized with isoflurane and stabilized on a stereotaxic apparatus (Kopf Instruments) for bilateral injections of recombinant AAV vectors into the prefrontal cortex (PFC) at Bregma coordinates 1.98 mm anterioposterior, ±0.5 mm mediolateral, and −2.35 mm dorsoventral. *Akt1* KO mice were administered AAV (1 μL at 2.5e12 GC/mL over 10 min) to express either human AKT1-GFP (Vector Biolabs, Malvern, PA) or Cre-GFP (Penn Vector Core, Philadelphia, PA) under the human synapsin I promoter (hSyn) in each hemisphere. WT mice received

only AAV-hSyn::Cre-GFP injections. Injectors were removed after an additional 10 min post-AAV infusion. Following standard post-operative care, mice remained in their home cages for 21 d post-surgery to allow AAV expression before fear memory and extinction testing. Following behavior tests, the brains from these mice were fixed and sectioned for microscopy using similar procedures described previously (*Levenga et al., 2017*) to confirm AAV expression in the PFC by GFP visualization and AKT1 immunostaining.

### Western blotting

Brain tissues were isolated from a subset of mice after completion of behavioral testing (*Figure 1—figure supplement 1*) and blotted using procedures described previously to probe phosphorylated and total levels of AKT isoforms, PDK1, GSK3β, and ERK (*Levenga et al., 2017*).

### Immunohistochemistry

Fluorescent immunostaining of fixed brain sections from transcardially perfused mice was performed using similar procedures described previously (*Levenga et al., 2017*) to visualize expression of AKT1, AKT3, NeuN, PV, and GAD67.

### Experimental design and statistical analysis

All data are presented as the mean ± SEM and were statistically evaluated using SPSS (IBM Corporation). Data were analyzed for each sex by Student's t-test, one-way analysis of variance (ANOVA) or one-way repeated-measures ANOVA where appropriate, with genotype as the between-subjects factor (*Supplementary file 1*). Homogeneity of variance was assessed using Levene's test. Outliers were excluded based on Grubbs' method, experimental criteria, or equipment errors (*Supplementary file 2*). *Akt* isoform mutants were maintained in separate lines and were not tested together, so *Akt* isoform experimental groups were compared only with their respective WT littermates and not analyzed together. Because male and female mice were tested separately, we also did not combine sexes for statistical analyses. However, we have included two-way ANOVA with sex and genotype as independent variables for each *Akt* mutant line (*Supplementary file 3*) showing that our findings using either statistical approach are largely unchanged. Experiments were designed based on power analyses derived from previously conducted similar experiments (*Hoeffer et al., 2008*; *Hoeffer et al., 2013*; *Wong et al., 2015*). All behavior experiments used a minimum of three independent litters derived from separate dams. Significant ANOVA results were followed by Tukey's HSD test for multiple group comparisons where appropriate. All statistical tests were two-tailed with $p < 0.05$ considered significant.

## Acknowledgements

These studies were supported by grants from the National Institutes of Health (R01 NS086933-01, R01 AG 064465, T32 MH016880 and T32 AG052371) and funds from the Linda Crnic Institute. We thank Nicole Kethley, Michael Roche, Daniel Peterson, Stephanie Quintana, and Emily Schmitt for technical contributions to this work and Drs. Luke Evans and Peter Cain for useful comments and discussions in the preparation of the manuscript. We dedicate this work in the loving memory of Ms. Lauren LaPlante, our friend and colleague whose contributions were not only critical to this study but also to the well-being of our research group.

## Additional information

### Funding

| Funder | Grant reference number | Author |
| --- | --- | --- |
| National Institute of Neurological Disorders and Stroke | NS086933 | Helen Wong<br>Josien Levenga<br>Lauren LaPlante<br>Bailey Keller<br>Andrew Cooper-Sansone<br>Curtis Borski<br>Ryan Milstead |

| | | Charles Hoeffer |
| --- | --- | --- |
| National Institute of Mental Health | MH016880 | Helen Wong Charles Hoeffer |
| Jerome Lejeune Foundation | 1805 | Helen Wong Andrew Cooper-Sansone Charles Hoeffer |
| National Institute on Aging | AG 064465 | Helen Wong Lauren LaPlante Charles Hoeffer |
| National Institute on Aging | AG052371 | Ryan Milstead |

The funders had no role in study design, data collection and interpretation, or the decision to submit the work for publication.

## Author contributions
Helen Wong, Conceptualization, Data curation, Formal analysis, Supervision, Validation, Investigation, Visualization, Methodology, Writing - original draft, Project administration, Writing - review and editing; Josien Levenga, Conceptualization, Data curation, Formal analysis, Supervision, Investigation, Methodology, Project administration, Writing - review and editing; Lauren LaPlante, Bailey Keller, Supervision, Investigation; Andrew Cooper-Sansone, Curtis Borski, Investigation; Ryan Milstead, Formal analysis, Investigation; Marissa Ehringer, Formal analysis, Methodology, Writing - review and editing; Charles Hoeffer, Conceptualization, Resources, Data curation, Formal analysis, Supervision, Funding acquisition, Validation, Investigation, Visualization, Methodology, Writing - original draft, Project administration, Writing - review and editing

## Author ORCIDs
Helen Wong https://orcid.org/0000-0002-3927-8953
Josien Levenga http://orcid.org/0000-0002-9971-6337
Ryan Milstead https://orcid.org/0000-0002-3333-853X
Charles Hoeffer https://orcid.org/0000-0002-2036-0201

## Decision letter and Author response
Decision letter https://doi.org/10.7554/eLife.56630.sa1
Author response https://doi.org/10.7554/eLife.56630.sa2

# Additional files

## Supplementary files
- Supplementary file 1. Statistical analysis using t-tests.
- Supplementary file 2. Outliers and justification.
- Supplementary file 3. Statistical analysis using ANOVAs.
- Transparent reporting form

## Data availability
All data generated or analyzed during this study are included in the manuscript and supporting files. Source data files have been provided for Figures 1-9.

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
