## [Decision Letter]

**Acceptance summary:**

The role of molecular isoforms, highly similar molecules, in behaviors relevant to psychiatric disorders are not well elucidated. Using a series of genetic and behavioral approaches this research uncovers a new role for sex-specific contributions of molecular isoforms in distinct behaviors related to memory, anxiety, or fear. These findings have implications for understanding molecular mechanisms relevant to biological sex that occur in neuropsychiatric related behaviors.

**Decision letter after peer review:**

Thank you for submitting your article "Isoform-specific roles for AKT in affective behavior, spatial memory, and extinction related to psychiatric disorders" for consideration by *eLife*. Your article has been reviewed by three peer reviewers, and the evaluation has been overseen by a Reviewing Editor and Kate Wassum as the Senior Editor. The reviewers have opted to remain anonymous.

The reviewers have discussed the reviews with one another and the Reviewing Editor has drafted this decision to help you prepare a revised submission.

As the editors have judged that your manuscript is of interest, but as described below that additional experiments are required before it is published, we would like to draw your attention to changes in our revision policy that we have made in response to COVID-19 (https://elifesciences.org/articles/57162). First, because many researchers have temporarily lost access to the labs, we will give authors as much time as they need to submit revised manuscripts. We are also offering, if you choose, to post the manuscript to bioRxiv (if it is not already there) along with this decision letter and a formal designation that the manuscript is 'in revision at *eLife*'. Please let us know if you would like to pursue this option. (If your work is more suitable for medRxiv, you will need to post the preprint yourself, as the mechanisms for us to do so are still in development.)

Summary:

This study examines AKT isoform-dependent effects on behaviors that are relevant to complex brain diseases. The study reveals differential selection of AKT1, AKT2, and AKT3 isoforms as a novel mechanism underlying anxiety, memory formation, and extinction. The authors further demonstrate that different AKT isoforms can compensate for each other, partially explaining the presence of very mild or null phenotypes. Overall, the manuscript expands upon the current limited knowledge of AKT signaling in maladaptive behaviors.

Essential revisions:

1) A concern with this study is that the results do not present a coherent conclusion, thus, it is not clear what is learned from this study. Please reshape the Introduction to clearly motivate the rationale to study these specific isoforms of AKT with genetic deletion, and what relevance this has for specific psychiatric disease and why the behavioral test performed are relevant. Additional comments below should also be integrated into this revised Introduction.

2) In the Introduction, and some parts of the Discussion, the authors indicate that AKT is associated with MDD, and that antidepressant medications, among other psychotropic agents, modify AKT signaling. Yet, the authors provide only one reference that supports such statement for antipsychotics only (Beaulieu et al., 2008). Please reference other sources to fully support such statements across the paper, particularly for antidepressants in human subjects (for example, Krishnan et al., 2008).

3) The NIH released a statement on the use of animals for research on psychiatric illness (https://grants.nih.gov/grants/guide/notice-files/NOT-MH-19-053.html). The statement emphasizes the importance of wording (including "of" versus "for") in descriptions and interpretations of the types of studies described in the present article. It would seem critical to ensure that the conceptualizations of genetically altered mice, and the behaviors they exhibit when compared to wildtypes, are made in the context of "modeling" endpoints that are suggestive of symptoms that underlie human psychiatric illnesses. Thus, it is imperative that the assertions in the present article are consistent with this NIH statement. As one example, the insinuation that the behaviors described (e.g., OFA, EPM, MWM, fear conditioning/extinction) can accurately assess neuropsychiatric behavior in a mouse (i.e., last paragraph of Introduction) may be viewed as increasingly unacceptable. Therefore, please revise to describe these tests in terms of their objective endpoints.

4) Following the previous point, at the end of the Introduction, the authors indicate that their "findings have significant implications for diagnoses and therapies." Please describe how this statement is consistent with the NIH considerations regarding the use of animal neurobehavioral approaches in basic studies.

5) One of prominent phenotypes in the study is a male-specific enhanced fear extinction in AKT1 KO. This can also be reversed by expressing AKT1 in the PFC. However, one of the main brain parts responsible for fear extinction is the amygdala and there are studies showing the connection between the PFC and amygdala may also play role in fear extinction. In addition, in Figure 6, conditional KO of AKT1 show normal extinction, which is possibly due to the fact that the cre line used in the study has not reported to have expression in the amygdala. This suggests AKT1 in the amygdala plays a role in fear extinction. Please examine whether AKT1 expression in the amygdala of KO mice can reverse fear extinction, or test whether specific loss of AKT1 in the amygdala (by viral shRNA knockdown in WT mice) has any effects on behavior.

6) The authors inject AKT1 into the Prelimbic cortex (PL), which is sufficient to reverse altered fear extinction in KO mice. Importantly, several studies show that the connection between the amygdala and PL mediates fear acquisition and retrieval while the link between the amygdala and infralimbic cortex (IL) are thought to mediate fear extinction (Choi et al., PNAS. 2010. and Laurent et al., Learning and Memory. 2009). Please explain how AKT1 expression in the PL can reverse fear extinction or examine the effects of AKT1 expression in the IL on fear extinction in KO animals.

Please also include a representative example of expression with sufficient display of anatomical landmarks to indicate that expression was restricted to the PL, as well as a schematic of expression spread overlaid for each subject so readers can see the extent of expression for the entire cohort.

7) Across the manuscript, the authors indicate that their findings are "sex-dependent." This is technically inaccurate, given that sex was not incorporated as an independent variable in their statistical analyses. This is particularly problematic in the Discussion section of the paper, since the experimental design does not support such inferences. A similar problem is exhibited for the arguments made across the different AKT isoforms (AKT1, AKT2, AKT3), since "isoform" was also not incorporated as an independent variable. The data should be re-analyzed using a 2-way ANOVA with "sex" and "AKT isoform" as sources of variance. Please also include a discussion on potential sex-dependent downstream players.

8) In the Experimental Design and Statistical analysis section of the paper, the authors indicate that outliers were excluded from the study. The specific number of excluded animals, across each experiment, should be clearly indicated. Also, the current manuscript only includes group numbers. The total number (n) of animals per experimental group needs to be included (a table depicting this information would be helpful, or adding the "n" within each group in the data figures). This is particularly important, given that the degrees of freedom change across experiments (this is confusing, since the authors indicate that mice were used across all experiments).

Because the numbers of animals differ considerable between some of the groups, please also show that the data meet the assumption of homogeneity of variance for ANOVA.

9) In Figure 6, please confirm loss of AKT1 expression by immunohistochemistry to examine the brain regions with AKT1 loss of expression.

10) Given the complexity and length of the behavioral tests, it would be very useful if the authors included a schematic representation of the experimental timeline, depicting the multiple groups of AKT mutants, the rescue experiment, and the timeline of the behavioral tests.

11) While the authors use a genetically targeted approach to study AKT isoforms, these molecules cross-talk with many signaling pathways. Therefore, it is very likely that multiple pathways are de-regulated by AKT KO. Assessing activation of other cross-talk signaling molecules (PI3-kinase, GSK3b, ERK, or CREB) across the multiple AKT mutants (Western blotting) would provide a better understanding of the role that each AKT isoform plays in the AKT-dependent behavioral modifications.

12) Some of the AKT KO mouse lines have previously been shown to be smaller and display growth deficiency (i.e., Dummler et al., 2006-DOI: 10.1128/MCB.00722-06; Cho et al., 2001-DOI: 10.1074/jbc.C100462200). Are the same lines used in the current study. Are the mice smaller? Did the authors check the general health state of the mice? It is suggested is to add a short paragraph in the manuscript that clarifies this point.

[Editors' note: further revisions were suggested prior to acceptance, as described below.]

Thank you for resubmitting your work entitled "Isoform-specific roles for AKT in affective behavior, spatial memory, and extinction related to psychiatric disorders" for further consideration by *eLife*. Your revised article has been evaluated by Kate Wassum (Senior Editor) and a Reviewing Editor.

The manuscript has been improved but there are some remaining issues that need to be addressed before acceptance, as outlined below:

1) The authors conclude that AKT1 in PFC interneurons is important for fear extinction. They showed that virally expressed AKT1 in the PFC reversed enhanced fear extinction in KO mice (Figure 5). However, the study lacks confirmation of AKT1 expression in interneurons. The authors are able to show double labeling of PV and AKT1 in the PFC in Figure 6—figure supplement 1. Thus, the authors should be able to perform similar labeling to demonstrate that virally expressed AKT1 is present in PFC interneurons in Figure 5C to confirm that AKT1 is indeed expressed in PFC interneurons of KO mice.

2) Describing sex as an "experimental variable" throughout the manuscripts is inaccurate. The authors are to be commended for including both males and females in their work, and indeed, the inclusion of both sexes is extremely important. But not including sex as an independent variable in their statistical analyses (ANOVAs), precludes the authors from suggesting that sex was an experimental variable (this is misleading, and technically incorrect). There is agreement with the justification provided by the authors for not including sex or isoform as a variable in their statistical approaches. For this reason, it is encouraged that the authors remove any statement indicating that their findings are isoform or sex "dependent", and simply rephrase them as sex and isoform-"specific" (because data is compared to a same sex control).

---

## [Author Response]

Essential revisions:1) A concern with this study is that the results do not present a coherent conclusion, thus, it is not clear what is learned from this study. Please reshape the Introduction to clearly motivate the rationale to study these specific isoforms of AKT with genetic deletion, and what relevance this has for specific psychiatric disease and why the behavioral test performed are relevant. Additional comments below should also be integrated into this revised Introduction.

To address the reviewers’ concerns on this important point we have revised the Introduction as well as the Significance Statement to reflect more clearly the goals and knowledge gaps that motivated these studies. AKT has been implicated in neuropsychiatric disorders from human studies but it is a central kinase in many cellular signaling pathways, so understanding the specificity of AKT activity will be important for improving diagnosis and therapeutic interventions. Our study investigated the isoform- and sex-specific functions of AKT in mouse behaviors with underlying neural circuits and activity that may model brain processes impacted in neuropsychiatric disorders. The use of genetic deletion is a standard approach for understanding the contributions of a gene toward a process of interest. While there are other approaches (e.g. gain of function, expression of allelic variants), these approaches are not currently available for mouse model work without additional reagent generation. We did use the approach of conditional cell type-specific *Akt* isoform removal and viral-mediated gene expression to extend our results beyond null characterization and feel that these studies are very informative above and beyond what is known about AKT function in the context of behavior and how these behaviors may relate to human neuropsychiatric disorder.

2) In the Introduction, and some parts of the Discussion, the authors indicate that AKT is associated with MDD, and that antidepressant medications, among other psychotropic agents, modify AKT signaling. Yet, the authors provide only one reference that supports such statement for antipsychotics only (Beaulieu et al., 2008). Please reference other sources to fully support such statements across the paper, particularly for antidepressants in human subjects (for example, Krishnan et al., 2008).

As suggested by the reviewers, we have included more references for AKT’s involvement in psychotropic agents, including antidepressants. These additional references add to the growing significance of the AKT signaling pathway in psychiatric disorders.

3) The NIH released a statement on the use of animals for research on psychiatric illness (https://grants.nih.gov/grants/guide/notice-files/NOT-MH-19-053.html). The statement emphasizes the importance of wording (including "of" versus "for") in descriptions and interpretations of the types of studies described in the present article. It would seem critical to ensure that the conceptualizations of genetically altered mice, and the behaviors they exhibit when compared to wildtypes, are made in the context of "modeling" endpoints that are suggestive of symptoms that underlie human psychiatric illnesses. Thus, it is imperative that the assertions in the present article are consistent with this NIH statement. As one example, the insinuation that the behaviors described (e.g., OFA, EPM, MWM, fear conditioning/extinction) can accurately assess neuropsychiatric behavior in a mouse (i.e., last paragraph of Introduction) may be viewed as increasingly unacceptable. Therefore, please revise to describe these tests in terms of their objective endpoints.

We understand that the behavioral measures we assessed in mice are endpoints which serve as models for examining effects on neural activity relevant to brain functions affected in psychiatric disorders and apologize for inadvertently using language that was not responsive to the NIH statement and guidance. We have revised the Introduction and wording in other places of the manuscript to be more objective and amenable to NOT-MH-19-053. The goals of our research were to identify behaviors and neurobiological processes impacted by AKT isoform activity and to examine them in the context of potential sex differences, which had been largely overlooked in previous preclinical studies. This has been more clearly delineated in the text.

4) Following the previous point, at the end of the Introduction, the authors indicate that their "findings have significant implications for diagnoses and therapies." Please describe how this statement is consistent with the NIH considerations regarding the use of animal neurobehavioral approaches in basic studies.

We have modified the Introduction to better fit the guidelines outlined in NOT-MH-19-053. In accordance with this notice, our study highlights the findings in the context of behavioral assays as readouts “for” brain function that may model neurobiological processes involved in psychiatric disorders. We delineate the contributions of AKT isoforms to specific brain processes (e.g. hippocampus-dependent long-term memory formation) and identify sex-dependent differences in the requirements for AKT activity.

5) One of prominent phenotypes in the study is a male-specific enhanced fear extinction in AKT1 KO. This can also be reversed by expressing AKT1 in the PFC. However, one of the main brain parts responsible for fear extinction is the amygdala and there are studies showing the connection between the PFC and amygdala may also play role in fear extinction. In addition, in Figure 6, conditional KO of AKT1 show normal extinction, which is possibly due to the fact that the cre line used in the study has not reported to have expression in the amygdala. This suggests AKT1 in the amygdala plays a role in fear extinction. Please examine whether AKT1 expression in the amygdala of KO mice can reverse fear extinction, or test whether specific loss of AKT1 in the amygdala (by viral shRNA knockdown in WT mice) has any effects on behavior.

The particular Cre line used for this study, T29-1 (Tsien et al., 1996), does in fact express quite well in the amygdala. In a previous report, we confirmed its expression in the amygdala (Hoeffer et al., 2008, Figure 1). To provide additional support, we have included immunoblotting and histological data confirming reduced AKT1 expression in the amygdala as well as in the PFC of conditional *Akt1* KO (*cAkt1* KO) mice driven by the T29-1 line (Figure 6—figure supplement 1). We show further that this line expresses in excitatory neurons and not interneurons of the amygdala and PFC (Figure 6—figure supplement 1). Taken together with our finding that neuronal AKT1 expression in the PFC of *Akt1* KO mice reverses their fear extinction phenotype, these data suggest that interneuronal AKT1 in the PFC plays a critical role in fear extinction. Therefore, *cAkt1* KO mice showed normal extinction possibly because AKT1 expression remained intact in interneurons of the PFC. We do not dispute the important role of the amygdala in extinction, but our data indicate that AKT1 is not required in excitatory neuronal populations of the amygdala or PFC for the normal display of extinction. To more thoroughly test this idea, we plan to generate interneuron-specific *Akt1* KO mice and examine fear extinction in these animals for future studies. We have included this discussion in the revised manuscript. Based on these data, we did not pursue behavioral studies to test the role of AKT1 specifically in the amygdala, although we agree that it would be interesting to examine as part of a larger future study of brain region-specific AKT1 functions.

6) The authors inject AKT1 into the Prelimbic cortex (PL), which is sufficient to reverse altered fear extinction in KO mice. Importantly, several studies show that the connection between the amygdala and PL mediates fear acquisition and retrieval while the link between the amygdala and infralimbic cortex (IL) are thought to mediate fear extinction (Choi et al., PNAS. 2010. and Laurent et al., Learning and Memory. 2009). Please explain how AKT1 expression in the PL can reverse fear extinction or examine the effects of AKT1 expression in the IL on fear extinction in KO animals.Please also include a representative example of expression with sufficient display of anatomical landmarks to indicate that expression was restricted to the PL, as well as a schematic of expression spread overlaid for each subject so readers can see the extent of expression for the entire cohort.

We apologize that the methods for viral-mediated AKT1 expression was not clearly described in the original submission. Although we targeted the PL region of the medial prefrontal cortex (mPFC), it was not our intention to imply that viral expression was limited to the PL. At the reviewers’ suggestion, we provide a representative example of AAV expression in the PFC along with a schematic of the expression spread overlaid for available subjects, which we have incorporated into Figure 5. While AAV expression was visually confirmed for every subject, we had only comprehensively imaged a few representative samples. As these experiments were performed over the course of a couple years, the original slides for confirming expression were no longer usable for imaging. We prepared new slides but only some subjects had brain sections with PFC available. To construct the schematic, we compiled images from this subset, which consists of samples from each group. The revised figure shows that both the PL and IL exhibit AAV expression (Figure 5B). We do not distinguish between the requirements for AKT1 activity in PL or IL in this study. Instead, we elected for a more general approach, indeed given some conflicting reports regarding the requirement for the PL and IL in fear extinction processes (Do-Monte et al., 2015; Sierra-Mercado et al., 2011; Marek et al., 2018). Our simplest interpretation of the results from restoring AKT1 expression in the mPFC of *Akt1* KO mice is that AKT1 activity is normally required in the mPFC to limit extinction learning. While examining the role of AKT1 activity in specific PFC regions would be an interesting line of inquiry, based on our results from forebrain excitatory neuron-specific removal of AKT1, we posit that the AKT1 activity utilized during extinction learning is primarily required in interneuronal populations of the PFC (Figure 6 and Figure 6—figure supplement 1). The AAV-AKT1 construct we used expresses in cells responsive to the synapsin promoter, so it does not resolve AKT1 activity in excitatory or inhibitory neurons (Figure 5). Therefore, as mentioned for point #5, our future efforts will directly test the idea that interneuronal AKT1 is responsible for normal extinction behavior using genetic approaches to eliminate AKT1 and AAV constructs designed to limit AKT1 expression to interneurons.

7) Across the manuscript, the authors indicate that their findings are "sex-dependent." This is technically inaccurate, given that sex was not incorporated as an independent variable in their statistical analyses. This is particularly problematic in the Discussion section of the paper, since the experimental design does not support such inferences. A similar problem is exhibited for the arguments made across the different AKT isoforms (AKT1, AKT2, AKT3), since "isoform" was also not incorporated as an independent variable. The data should be re-analyzed using a 2-way ANOVA with "sex" and "AKT isoform" as sources of variance. Please also include a discussion on potential sex-dependent downstream players.

This is a very important point, and we carefully considered the implications of our research design and interpretations to address this critique. It appears to us that many labs employ a variety of statistical approaches, some applied correctly, some applied incorrectly. It is also the case that there are often multiple statistical approaches that could be used, and if the “take-home” message doesn’t change it can simply be a matter of preference. The two main issues raised by reviewers for this manuscript focus on how to handle possible sex differences and whether to examine possible *Akt* isoform differences. Below we present our rationale for the way we conducted the analyses.

1) Sex differences: We do not share the reviewers’ interpretation about the use of “sex-dependent.” Behavioral effects observed in one sex but not another is sex-dependent. However, we understand the concept of including sex as a variable in our statistical analyses. In a previous submission to another journal that included only a subset of these data, we did in fact prepare our manuscript with two-way ANOVA analyses (genotype and sex) but received criticisms about that approach. In that case, the reviewers requested post-hoc analyses, but post-hoc tests are not appropriate when there are only two levels for each variable (e.g. genotype can be only KO/WT and sex can be only M/F). Therefore, we consulted with local biostatistical experts (Dr. Luke Evans and Dr. Marissa Ehringer) and revised our figures based on their directions.

Dr. Ehringer deals with this sex effect issue on a regular basis because there are well-known sex differences in both rodents and humans for alcohol-related behaviors, which is a major area of her research. Given the *a priori* knowledge of such sex differences (Choleris et al., 2018; Andreano and Cahill, 2009; Wiltgen et al., 2005; File, 2001), analyses are frequently performed separately by sex to simplify the presentation and interpretation of results. It is not uncommon for researchers to actually submit two different papers, one for each sex. In our case, the behaviors we examined are also well-known to show reproducible sex differences in WT (C57BL/6) strains (Archer, 1977; Wiltgen et al., 2001; Bolivar et al., 2001; Voikar et al., 2001; La Buda et al., 2002; Gresack and Frick, 2003; Guo et al., 2004; Berger-Sweeney et al., 2005; Jonasson, 2005; Calderone et al., 2008; An et al., 2011; Davis et al., 2012; Matsuda et al., 2015; Arakawa, 2019). Thus, to avoid complicating the current submission with results of behavioral differences inherent to sex alone, which are already known, and because males and females within each genotype set were tested separately, we opted to perform within-sex t-tests (except for Figure 5, in which the independent variable had three levels). This analysis facilitated presentation of our results for easy comparison between WT and KO genotypes within either male or female groups. It allowed us to show data normalized to WT controls within each sex in order to focus on the effect of *Akt* isoform manipulation, which was the primary goal of our study. At the reviewers’ suggestion, we also performed two-way ANOVA, and the overall “take-home” message does not change. This analysis is included in Table 4. If all reviewers and the editor strongly feel we should present the results as a two-way ANOVA, we can change our figures. Based on our previous submission experience, we expect this might cause more confusion than the simpler t-test, which we believe is justified based on previously known sex differences and the experimental design.

2) AKT isoform comparisons: We believe including *Akt* isoform as an independent variable for analyses is not appropriate for four reasons. First, the individual *Akt* mutant lines are bred and maintained separately. They cannot be reasonably grouped all together because they are maintained on different backgrounds and not simply interchangeable. In order to analyze the data as proposed (all *Akt* mutants simultaneously), all the mutant *Akt* alleles would have to be maintained in the same strain. This was not done for practical reasons; it would require thousands of progeny from *Akt1***^+/-^***Akt2***^+/-^***Akt3***^+/-^** breeder mice to complete these studies in this way and only 1.6% of mice derived from such a cross would be either pure WT or KO for each isoform group. Second, although all the mice are maintained on a C57Bl/6 background, the different *Akt* lines are not isogenic. We regularly introduce new C57Bl/6 mice into all our strains to prevent the accumulation of genetic modifiers. Despite this precaution, “WT” mice from each strain demonstrate slight performance differences in some tasks compared with “WT” mice from other *Akt* isoform strains. There are many possible reasons for this (genetics, rearing, microbiome differences, etc.) but those factors are largely beyond our control. Therefore, it would not be scientifically appropriate to compare KO mice to “WT” mice from a different line or to create a master “WT” group by simply averaging the performance of WT mice from all lines. Third, we used littermates obtained from breeding the individual *Akt* lines for each experimental group, a practice that is supported by previous studies (Holmdahl and Malissen, 2012). This important element of rigorous experimental design would be lost if we were to combine across all isoform groups. Finally, the experiments in this study were not designed or performed with the goal of comparing each *Akt* mutant line to one other. Combining all the mutants for simultaneous comparison creates an insurmountable multiple comparison problem that would necessitate testing many more animals, as the power analyses used to design our experiments were based on separate isoform testing. Another way to view the situation is that because the experiments were performed with only one mutant at a time, it would have been reasonable and appropriate to prepare three different manuscripts, one for each isoform, which would be analyzed independently.

At the reviewers’ request, we also included in the revised manuscript a discussion on potential sex-dependent downstream players, which incorporates the findings from our experiments to address point #11 below.

8) In the Experimental Design and Statistical analysis section of the paper, the authors indicate that outliers were excluded from the study. The specific number of excluded animals, across each experiment, should be clearly indicated. Also, the current manuscript only includes group numbers. The total number (n) of animals per experimental group needs to be included (a table depicting this information would be helpful, or adding the "n" within each group in the data figures). This is particularly important, given that the degrees of freedom change across experiments (this is confusing, since the authors indicate that mice were used across all experiments).Because the numbers of animals differ considerable between some of the groups, please also show that the data meet the assumption of homogeneity of variance for ANOVA.

We apologize that presentation of the experimental details for our statistical analysis was confusing in the original submission. We have now added summary tables with the number of animals per experimental group (also included in the figure legends), number of excluded animals and reason for exclusion, and homogeneity of variance test results across each experiment (Tables 2-4).

9) In Figure 6, please confirm loss of AKT1 expression by immunohistochemistry to examine the brain regions with AKT1 loss of expression.

We previously confirmed that conditional *Akt1* removal under the T29-1 *Camk2α-*Cre driver reduced AKT1 expression in the hippocampus (Levenga et al., 2017). Complementing this report, we now demonstrate using western blotting and immunohistochemistry that AKT1 expression is also reduced in the amygdala and PFC. These data have been added as a supplemental figure (Figure 6—figure supplement 1).

10) Given the complexity and length of the behavioral tests, it would be very useful if the authors included a schematic representation of the experimental timeline, depicting the multiple groups of AKT mutants, the rescue experiment, and the timeline of the behavioral tests.

We thank the reviewers for this very useful suggestion and have amended our Materials and methods section to include a flowchart for the behavioral assays (Figure 1—figure supplement 1).

11) While the authors use a genetically targeted approach to study AKT isoforms, these molecules cross-talk with many signaling pathways. Therefore, it is very likely that multiple pathways are de-regulated by AKT KO. Assessing activation of other cross-talk signaling molecules (PI3-kinase, GSK3b, ERK, or CREB) across the multiple AKT mutants (Western blotting) would provide a better understanding of the role that each AKT isoform plays in the AKT-dependent behavioral modifications.

We agree that loss of AKT likely influences many different signaling pathways and that these may be involved in the behavioral effects we observed. Following the reviewers’ suggestion, we performed western blot analysis of PDK1, GSK3β, and ERK activation to assess candidate signaling upstream, downstream, and parallel of AKT, respectively, in brain tissue from *Akt* isoform mutants of both sexes. These data are now included as Figure 9 and associated supplemental figures. While we did not identify many differences in this analysis, we did find some exciting isoform- and sex-dependent effects on GSK3β phosphorylation. We hope to investigate other potential signaling differences that may mediate the behavioral effects of *Akt* isoform deficiency using a more comprehensive approach, such as mass spectrometry, in a follow-up study.

12) Some of the AKT KO mouse lines have previously been shown to be smaller and display growth deficiency (i.e., Dummler et al., 2006-DOI: 10.1128/MCB.00722-06; Cho et al., 2001-DOI: 10.1074/jbc.C100462200). Are the same lines used in the current study. Are the mice smaller? Did the authors check the general health state of the mice? It is suggested is to add a short paragraph in the manuscript that clarifies this point.

We obtained our *Akt1* and *Akt2* KO lines from JAX, donated by Morris Birnbaum. Our *Akt3* KO line was also obtained from the Birnbaum Lab by direct request and is not the same strain described in Dummler et al. 2006. This information has been added to the Key Resources Table. In our hands, both *Akt1* and *Akt3* KO mice show significantly reduced sizes compared with their WT littermates. The size reduction is slight in *Akt1* KO mice and more pronounced in *Akt3* KO mice, consistent with previous reports (Cho et al., 2001a (PMID: 11533044); Easton et al., 2005). There are conflicting reports of *Akt2* KO mice having a reduced weight (Cho et al., 2001b (PMID: 11387480); Garofalo et al., 2003). In our hands, *Akt2* KO mice trend towards a reduced size but the difference is not significant. *Akt2* KO mice do develop insulinemia that becomes progressively worse with age (Cho et al., 2001b; Garofalo et al., 2003). Most glucose-related phenotypes in *Akt2* KO mice are uncovered following fasting or with aging, but our mice have access to food *ad libitum* and were tested younger than 16 weeks, before insulin-related phenotypes become severe. Previous studies examined the general health of all these KO mice, and apart from size, found them generally indistinguishable from their WT littermates (Cho et al., 2001a; Cho et al., 2001b; Easton et al., 2005). In addition, we assessed sensory functions (acoustic startle, vision, and pain perception) and found no differences between KO and WT mice. Locomotor assessments, such as distance moved and swim velocity, were included in our behavioral assays to rule out locomotor defects as a potential confound for observed behavioral differences. None were identified. Some of these control data are included in the submitted figures (e.g. distance moved, visual platform performance, baseline freezing). We have added a paragraph about general health to the Materials and methods section to make clear this point.

[Editors' note: further revisions were suggested prior to acceptance, as described below.]

The manuscript has been improved but there are some remaining issues that need to be addressed before acceptance, as outlined below:1) The authors conclude that AKT1 in PFC interneurons is important for fear extinction. They showed that virally expressed AKT1 in the PFC reversed enhanced fear extinction in KO mice (Figure 5). However, the study lacks confirmation of AKT1 expression in interneurons. The authors are able to show double labeling of PV and AKT1 in the PFC in Figure 6—figure supplement 1. Thus, the authors should be able to perform similar labeling to demonstrate that virally expressed AKT1 is present in PFC interneurons in Figure 5C to confirm that AKT1 is indeed expressed in PFC interneurons of KO mice.

As suggested, we now include images confirming that virally expressed AKT1 in the PFC of *Akt1* KO mice is present in interneurons with our AAV construct (Figure 5—figure supplement 1).

2) Describing sex as an "experimental variable" throughout the manuscripts is inaccurate. The authors are to be commended for including both males and females in their work, and indeed, the inclusion of both sexes is extremely important. But not including sex as an independent variable in their statistical analyses (ANOVAs), precludes the authors from suggesting that sex was an experimental variable (this is misleading, and technically incorrect). There is agreement with the justification provided by the authors for not including sex or isoform as a variable in their statistical approaches. For this reason, it is encouraged that the authors remove any statement indicating that their findings are isoform or sex "dependent", and simply rephrase them as sex and isoform-"specific" (because data is compared to a same sex control).

We have changed references of “dependent” to “specific” throughout the text.